# A Lagrangian Perspective on the Lifecycle and Cloud Radiative Effect of Deep Convective Clouds Over Africa

William K. Jones[1], Martin Stengel[2], and Philip Stier[1]

[1]Atmospheric, Oceanic and Planetary Physics, Department of Physics, University of Oxford
[2]Deutscher Wetterdienst (DWD)

**Correspondence:** William K. Jones (william.jones@physics.ox.ac.uk)

**Abstract.** The anvil clouds of tropical deep convection have large radiative effects in both the shortwave (SW) and longwave (LW) spectra with the average magnitudes of both over $100\,\mathrm{Wm^{-2}}$. Despite this, due to the opposite sign of these fluxes, the net average of anvil cloud radiative effect (CRE) over the tropics is observed to be neutral. Research into the response of anvil CRE to climate change has primarily focused on the feedbacks of anvil cloud height and anvil cloud area, in particular regarding the
LW feedback. However, tropical deep convection over land has a strong diurnal cycle which may couple with the shortwave component of anvil cloud radiative effect. As this diurnal cycle is poorly represented in climate models it is vital to gain a better understanding of how its changes impact anvil CRE.

To study the connection between deep convective cloud (DCC) lifecycle and CRE, we investigate the behaviour of both isolated and organised DCCs in a 4-month case study over sub-Saharan Africa (May-August 2016). Using a novel cloud
tracking algorithm, we detect and track growing convective cores and their associated anvil clouds using geostationary satellite observations from Meteosat SEVIRI. Retrieved cloud properties and derived broadband radiative fluxes are provided by the CC4CL algorithm. By collecting the cloud properties of the tracked DCCs, we produce a dataset of anvil cloud properties along their lifetimes. While the majority of DCCs tracked in this dataset are isolated, with only a single core, the overall coverage of anvil clouds is dominated by those of clustered, multi-core anvils due to their larger areas and lifetimes.

We find that the anvil cloud CRE of our tracked DCCs have a bimodal distribution. The interaction between the lifecycles of DCCs and the diurnal cycle of insolation results in a wide range of SW anvil CRE, while the LW component remains in a comparatively narrow range of values. The CRE of individual anvil clouds varies widely, with isolated DCCs tending to have large negative or positive CREs while larger, organised systems tend to have CRE closer to zero. Despite this, we find that the net anvil cloud CRE across all tracked DCCs is close to neutral ($-0.94 \pm 0.91\,\mathrm{Wm^{-2}}$). Changes in the lifecycle of DCCs, such
as shifts in the time of triggering, or the length of the dissipating phase, could have large impacts on the SW anvil CRE and lead to complex responses that are not considered by theories of LW anvil CRE feedbacks.

# 1 Introduction

Deep Convective Clouds (DCCs) play a key role in the tropical atmosphere. Forming the ascending branch of the Hadley cells near the equator, DCCs are critical to the circulation and heat transfer of the tropics (Riehl and Malkus, 1958; Weisman, 2015). DCCs are also a cause of extreme weather events including floods, lightning and hail (Westra et al., 2014). Mesoscale Convective Systems (MCSs)—large, long-lived convective systems in which the anvils of multiple convective cores combine into a single, large 'cloud shield' (Chen and Houze Jr, 1997; Houze, 2004; Roca et al., 2017)—are responsible for the majority of precipitation in the tropics (Feng et al., 2021). Changes in the behaviour of DCCs with climate change have the potential for major impacts on the atmosphere, weather and society.

DCCs also exert a key influence on the temperature of the tropics through their Cloud Radiative Effect (CRE). Due to their size, height and depth, DCC anvils have large radiative effects in both the Shortwave (SW) and Longwave (LW), with both having average magnitudes exceeding $100 \, \mathrm{Wm^{-2}}$ (Hartmann, 2016; Wall and Hartmann, 2018). However, due to the opposite signs of these two components, the average anvil CRE in the tropics is approximately zero (Ramanathan et al., 1989; Hartmann et al., 1992; Stephens et al., 2018). Radiation is also key to the lifecycle of DCCs. Over land, convection is typically initiated by the heating of the surface and lower troposphere by solar SW radiation, resulting in a peak of convective activity in the late afternoon. Over the ocean, however, convection is often triggered by LW cooling of the upper troposphere, and so convective activity occurs more frequently in the morning. However, the occurrence of convection is more uniform throughout the diurnal cycle compared to that over land (Taylor et al., 2017). Radiation also has an impact on DCC lifecycle through the differential heating of the anvil cloud, which destabilises the anvil cloud leading to dissipation due to entrainment and evaporation. However, SW heating of the anvil cloud top during daytime acts to stabilise and delay this process, leading to differences in anvil lifetime depending on the diurnal cycle (Harrop and Hartmann, 2016; Sokol and Hartmann, 2020; Wall et al., 2020).

There are several hypotheses regarding the CRE of tropical anvil clouds that consider whether the neutral CRE of tropical anvils is the result of a feedback mechanism. Ramanathan et al. (1989) proposed the thermostat hypothesis in which, in response to a warming environment, anvil clouds produce thicker cirrus which acts to cool the tropics through increased SW reflectance. The Iris hypothesis proposes that anvil cirrus will decrease in area, resulting in greater LW emission from the surrounding clear-sky regions. Lindzen et al. (2001) first proposed this as a result of increased precipitation efficiency, however evidence for this effect is disputed (Del Genio and Kovari, 2002; Lin et al., 2004). Bony et al. (2016) proposed a 'stability iris' feedback, in which the established trends of increased dry static stability (Held and Soden, 2006) and a reduction in the tropical overturning circulation (Vecchi and Soden, 2007) reduce the detrainment of anvil cirrus. Although the anvil cloud response is generally considered to have a negative climate feedback, the predicted magnitude varies widely, it represents the greatest uncertainty among all cloud feedbacks (Sherwood et al., 2020), and a positive feedback cannot be ruled out (Gasparini et al., 2023).

On the other hand, the Fixed Anvil Temperature (FAT) hypothesis argues that the anvil Cloud Top Temperature (CTT) remains constant in a warming climate, and the greater difference between anvil and surface temperature results in a positive LW feedback (Hartmann and Larson, 2002). The basis for FAT is that LW cooling of the troposphere due to water vapour

becomes inefficient below 220 K (Jeevanjee and Fueglistaler, 2020), which, if relative humidity remains constant, fixes the top of the convectively active troposphere at this isotherm. While there is evidence that this is the case for the largest DCC anvils, the increase in static stability may result in a reduced positive feedback due to a 'proportionally higher' anvil temperature (Zelinka and Hartmann, 2010) which more closely matches the LW response of tropical clouds in global climate models. While satellite observations have shown a trend in anvil cloud height (Norris et al., 2016), there is not yet sufficient evidence to distinguish this from inter-annual variability (Takahashi et al., 2019). Seeley et al. (2019), argued that FAT is a weak constraint on anvil temperature as while the radiative tropopause temperature remains fixed, the temperature of the tropopause lapse rate inversion can vary widely. Furthermore, as anvils tend to detrain below the tropopause, (Takahashi et al., 2017; Wang et al., 2020), anvil temperature and the tropopause temperature may only be weakly connected. Seidel and Yang (2022) however found the inclusion of $CO_2$ radiative heating produces anvil temperatures consistent with FAT.

While the iris and FAT feedbacks may act to cancel each other out, and hence maintain the neutral CRE of tropical anvil clouds, other potential feedback mechanisms may influence this balance. Hill et al. (2023) showed recently that climate models underestimate dynamically driven cloud feedbacks. Furthermore, convective instability is expected to scale with temperature in the same manner as the Clausius-Clapeyron relation (Seeley and Romps, 2015; Agard and Emanuel, 2017), and some observations of tropical anvil clouds have instead suggested that warming of the surface invigorates convection (Igel et al., 2014). Multi-decadal satellite observations have shown a cooling of upper tropospheric cloud temperature over land (Liu et al., 2023), indicating that changes in convective processes may lead to stronger cooling feedbacks.

Changes to the lifecycle and diurnal cycle of deep convection may also be an important factor, particularly when considering the SW feedback. Nowicki and Merchant (2004) used estimates of Top-of-Atmosphere (ToA) LW and SW radiative fluxes from Spinning Enhanced Visible Infra-Red Imager (SEVIRI) observations to estimate the diurnal cycle of anvil CRE over equatorial Africa and the equatorial Atlantic. They found that shifting the diurnal cycle of deep convection in these regions could change the CRE by $\pm 10\,\mathrm{W m^{-2}}$, but did not track the properties of individual DCCs. Bouniol et al. (2016) compared CRE and cloud radiative heating rates to anvil cloud properties to investigate how radiative heating affects the anvil cloud evolution. These observations were made with polar orbiting instruments however, and they highlighted the need for geostationary observations to characterise the evolution of individual anvil clouds. Subsequent research used DCC tracking methods to better characterise the lifecycle of observed anvil clouds (Bouniol et al., 2021), but as the radiative flux data was provided by polar-orbiting satellites the CRE could not be measured over the lifetime of the DCC.

In this article, we use a novel cloud tracking methodology in conjunction with derived all-sky and clear-sky radiative fluxes to characterise the CRE over the lifecycles of individual anvil clouds. This methodology is applied to 4 months of data produced for the ESA Cloud-CCI+ project over sub-Saharan Africa. This dataset allows us to investigate both the CRE of individual DCCs, as well as the net anvil CRE over the entire region. We find that the overall distribution of anvil CRE is determined by the relationship between DCC lifecycle and the diurnal cycle of the SW CRE, and discuss the implications of this for the response of DCCs to a changing climate.

**Table 1.** SEVIRI channels and their use in the DCC tracking algorithm and cloud properties retrieval.

| Channel | Wavelength (µm) | Description | Tracking | Retrieval |
|---------|-----------------|-------------|----------|-----------|
| 1 | 0.64 | Visible | | ✓ |
| 2 | 0.81 | NIR | | ✓ |
| 3 | 1.64 | NIR | | ✓ |
| 4 | 3.92 | NIR Window | | ✓ |
| 5 | 6.25 | Upper troposphere Water Vapour (WV) | ✓ | ✓ |
| 6 | 7.35 | Lower troposphere WV | ✓ | ✓ |
| 7 | 8.70 | Mid-IR window | | |
| 8 | 9.66 | Ozone | | |
| 9 | 10.8 | Clean LW window | ✓ | ✓ |
| 10 | 12.0 | Dirty LW window | ✓ | ✓ |
| 11 | 13.4 | $CO_2$ | | ✓ |
| 12 | 0.6–0.9 | High-resolution visible | | |

## 2 Data

For this case study, we used data from SEVIRI (Aminou, 2002) aboard the Meteosat Second Generation Meteosat-11 satellite, which is in a geostationary orbit above the equator at 0°W. We use data from 4 months (May–August 2016) over sub-Saharan Africa (approximately 18 °W–46 °E, 31 °S–15 °N) at the full resolution of SEVIRI (3 km at nadir) as well as retrieved cloud properties and derived broadband fluxes produced by the ESA Cloud-CCI+ project. Brightness Temperature (BT) from SEVIRI is used by the tracking algorithm, and reflectances and BT are used by the cloud retrieval.

SEVIRI is a visible and Infrared radiometer with a nadir spatial resolution of 3 km and a temporal sampling time of 15 minutes for the full earth disc. SEVIRI has 12 channels across the visible, Near Infrared (NIR) and thermal-IR spectrum, with one being a high-resolution visible channel with a nadir resolution of 1 km. A brief overview of these channels, along with which are used for tracking DCCs and the cloud properties retrieval, is provided in table 1.

An example of observations from SEVIRI is shown in fig. 1 for 15:00:00 UTC on 1[st] June 2016. A visible composite (fig. 1 a) is constructed using the 1.64 µm and 0.81 µm near-infrared and 0.64 µm visible channels for the Red, Green, Blue (RGB) channels respectively. In this composite, ice clouds (which appear cyan) can be seen over central Africa and the southern Atlantic. fig. 1 b shows the 10.8 µm brightness temperature for the same scene, showing the coldest temperatures for the high ice clouds over central Africa. Two combinations of channels are used for the detection of anvil clouds. The Water Vapour Difference (WVD), shown in fig. 1 c, consists of the 6.3 µm BT minus the 7.4 µm BT. In clear skies the WVD is negative, with values around $-20$ to $-15$ K, due to the higher, and thus colder, emission height of the 6.3 µm channel. In high, thick clouds, however, the temperatures of the 6.3 and 7.4 µm channels converge and so the WVD becomes closer to 0. In the cases

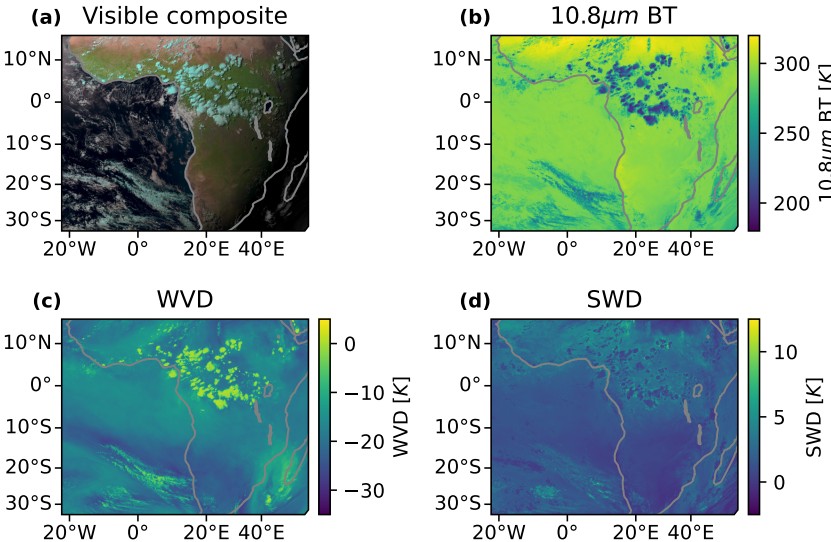

**Figure 1.** Example observations from the Meteosat SEVIRI instrument at 15:00:00 UTC on 2016/6/01. a: A visible composite formed using the 1.6, 0.81 and 0.64 μm channels as the RGB channels respectively, with 10.8 μm BT during the night-time. The scene shows a cluster of cold cloud tops (cyan) over central Africa and over the Southern Atlantic. b: 10.8 μm BT. c: WVD formed by the 6.3 μm channel minus the 7.4 μm channel. d: SWD formed by the 10.8 μm channel minus the 12.0 μm channel.

of the highest clouds, the WVD can become positive due to emission from stratospheric WV in the 6.3 μm channel. The Split Window Difference (SWD), shown in fig. 1 d, consists of the 10.8 μm BT channel minus the 12.0 μm channel. While the SWD is sensitive to near-surface WV due to absorption in the 12.0 μm channel, it is also sensitive to thin ice clouds due to the difference in emissivity of ice particles between the two channels. While for thick clouds the SWD will be 0 K, for thin ice clouds the lower emission height of the 10.8 μm BT channel results in a positive value of 5 K. The 10.8 μm and 12.0 μm channels of SEVIRI have relatively wide wavebands and as such are less sensitive to the presence of thin ice clouds. As a result, we found that the detection of thin anvil is unreliable using this technique with SEVIRI, and so is not considered within this article.

Retrieved cloud properties, including optical thickness, effective radius, liquid/ice water path, CTT and height, are provided by the Community Cloud Retrieval for Climate (CC4CL) algorithm (Sus et al., 2018; McGarragh et al., 2018). These properties are all retrieved at the same resolution as the input SEVIRI data. Broadband fluxes are derived using the BUGSRad radiative transfer model (Stephens et al., 2001) using input cloud properties from the CC4CL retrieval and vertical temperature, moisture and trace gas profiles from ERA-5 (Hersbach et al., 2020). The BUGSRad model provides ToA and Bottom-of-Atmosphere LW and SW radiative fluxes for both all-sky and clear-sky conditions. An example of these derived fluxes is shown in fig. 2. Figure 2 a shows net ToA fluxes, with a net warming during the daytime on the Western side of the image, and a net cooling at night-time on the Eastern side. Figure 2 b shows the net ToA CRE, with a net cooling effect during the daytime and warming

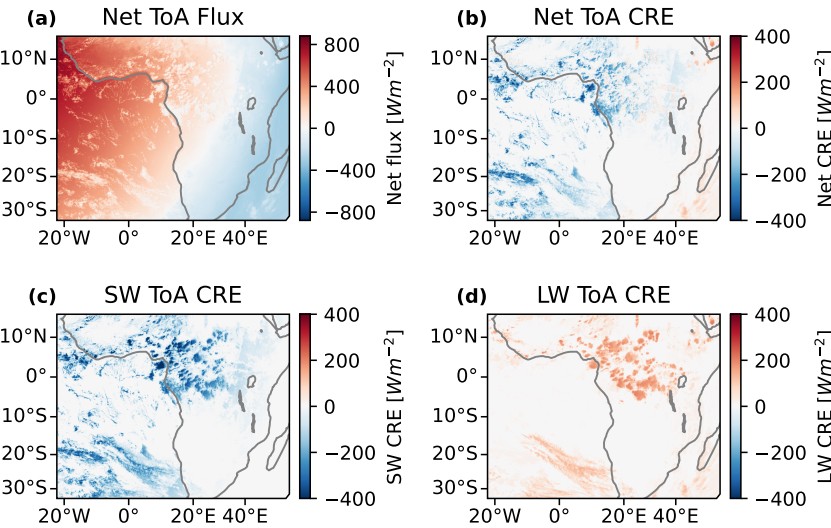

**Figure 2.** An example of the ToA CRE derived using the radiative flux model, for the same time as shown in fig. 1 (15:00:00 UTC on 2016/6/01). a: net ToA radiative flux. b: net ToA CRE. c: SW downwards CRE. d: LW downwards CRE.

during the night-time for observed high clouds over central Africa. The SW (fig. 2 c) and LW (fig. 2 d) components of the CRE show that while the LW, warming component has a smaller magnitude than the day-time, cooling SW CRE, it remains constant during both day- and night-time.

Validation of the SEVIRI broadband fluxes was performed against monthly-mean observations of ToA broadband CRE from the Clouds and the Earth's Radiant Energy System (CERES) (Loeb et al., 2018) Energy Balanced and Filled (EBAF) climate data record. The results of this validation are shown in fig. 3. Monthly mean fluxes were calculated for SEVIRI by first calculating the mean daily fluxes over each 1×1°grid square for days in which we have over 23 hours of observations, and then averaging these daily means over each month. Comparison of the net ToA CRE to CERES revealed a bias of $-1.87\,\mathrm{Wm^{-2}}$ (fig. 3 a,b), consisting of a SW bias of $-2.02\,\mathrm{Wm^{-2}}$ (fig. 3 c,d) and a LW bias of $+0.15\,\mathrm{Wm^{-2}}$ (Fig 3 e,f). Correction for these biases have been applied uniformly to all further CRE values given in this article.

## 3 Method

The detection and tracking of DCCs was performed using the tobac-flow algorithm (Jones et al., 2023), which has been designed specifically to track both isolated and clustered DCCs in geostationary satellite imagery over their entire lifecycle. While geostationary satellite imagery provides high-resolution observations over large domains and long time periods, which is ideal for studying deep convection, the inability of passive remote sensing to observe convective updrafts directly makes the detection and tracking of DCCs difficult.

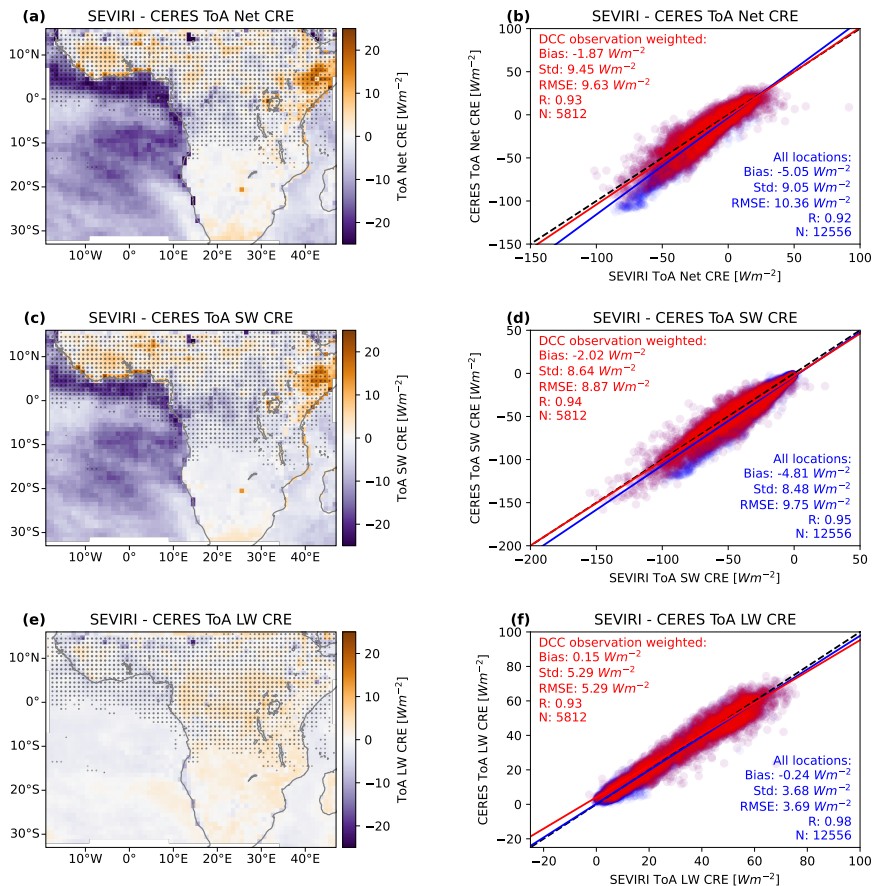

**Figure 3.** Validation of derived broadband fluxes against monthly CERES-EBAF CRE. a.: The mean difference in net ToA CRE by 1×1°grid square. b.: A comparison of observed ToA net CRE for SEVIRI against CERES, with all locations in blue, and those where we observe DCC anvils in red. c.: the mean difference in SW ToA CRE. d.: comparison of SW ToA CRE for SEVIRI and CERES. e.: the mean difference in LW CRE. f.: comparison of LW ToA CRE. The stippling in a, c and e represents the locations in which we observe DCC anvils, with the size of the dots corresponding to the number of observations. The solid lines in b, d and f show the linear regression for all locations (blue) and the locations where we observe DCC anvils (red) weighted by the number of observations.

Algorithms for the detection and tracking of DCCs in satellite imagery have generally been developed for one of two applications: tracking convective cores and isolated convection, or tracking large MCS anvils. Those designed for tracking deep convective cores, or isolated DCCs, include Cb-TRAM (Zinner et al., 2008, 2013) or tobac (Heikenfeld et al., 2019). These algorithms work by detecting regions of convective updraft or a proxy (such as cloud top cooling rate), and then treating these regions as point-like objects that are advected over time. The second group, designed for tracking MCSs, include algorithms

such as PyFLEXTRKR (Feng et al., 2022), TAMS (Núñez Ocasio et al., 2020) or TOOCAN (Fiolleau and Roca, 2013). These algorithms detect large regions of cold cloud tops which indicate anvil clouds, and then link them over time by overlapping regions at subsequent time steps. There is no 'best' method for tracking all types of convection however (Lakshmanan and Smith, 2010). The algorithms for tracking isolated convective cells perform worse for clustered convection when the motion and shape of the DCC cannot be adequately represented as a single vector. On the other hand, the MCS tracking algorithms

perform worse for smaller, isolated DCCs as the motion of the anvil between time steps may mean it does not overlap with the previous step.

    To approach the challenge of tracking both isolated DCCs and large, clustered systems, we address the role of cloud motion in the scaling problem. tobac-flow first estimates the motion of DCCs at each pixel using an optical-flow algorithm. Then, using these estimated motion vectors, we construct a semi-Lagrangian framework in which to perform the detection and tracking.

This approach addresses two issues found in traditional cloud tracking approaches. First, estimating a motion vector for each pixel allows complex motions (including divergence, rotation, splitting and merging) to be compensated for, rather than just the bulk motion found using the centroid tracking methods. Second, by estimating the cloud motion *a priori*, we are able to use this information within the detection step of the algorithm, and can separate changes in cloud properties such as BT from those observed due to cloud motion. This framework removes the problem of DCC motion, allowing us to track both isolated

and large DCCs at the same time.

    Three channels and channel combinations from SEVIRI are used for the detection algorithm: the $10.8\,\mu m$ BT channel; the WVD (the difference between the $6.2\,\mu m$ and $7.3\,\mu m$ channels), and the SWD (the difference between the $10.8\,\mu m$ and $12.0\,\mu m$ channels). Estimation of cloud motion vectors using optical flow in performed using the $10.8\,\mu m$ BT channel. Detection of cores utilises the $10.8\,\mu m$ BT channel and the WVD, and detection of anvils uses WVD and SWD.

We detect growing convective cores where we observe regions of rapid cooling in the $10.8\,\mu m$ BT channel of greater than $0.5\,Ks^{-1}$ and the WVD of greater than $0.25\,Ks^{-1}$. In both cases DCCs close to the surface and continue tracking them into the upper troposphere. We classify a core as a region of cooling temperature that has existed for at least 15 minutes and has cooled by at least $8\,K$ in a 15-minute period. This threshold provides a strong indicator of intense convective activity (Roberts and Rutledge, 2003), and so provides an accurate detection of growing DCCs.

Starting from these convective cores, we then detect the surrounding anvil cloud using the WVD field (Müller et al., 2018, 2019) and continue to detect the anvil until its dissipation, even after the core is no longer visible. A core and anvil are linked with each other if the two overlap at a time when the core has mean WVD of $> 5\,K$, indicating that it has reached the upper troposphere. Each anvil cloud can be associated with multiple cores, allowing us to identify cases of clustered convection. As we detect the cores based on cloud-top cooling, however, we can only detect the cores themselves during the growing phase,

and cannot detect cores that occur underneath cold, high, anvil clouds. When determining the number of cores associated with an anvil, we count the total number of cores observed over the entire lifetime, even if they do not occur at the same time, and including all merges and splits of the anvil cloud. This flexibility allows for a wide range of different organised systems to be analysed.

Due to the lack of sensitivity of the SEVIRI SWD to thin ice clouds, we only detect and track the thick portion of the anvil
in this article. The WVD channel of SEVIRI is capable of detecting anvils with optical thicknesses of approximately 1–1.5 (see supplementary fig. S1). However, the closer spacing and narrower bandwidth of the SEVIRI LW window channels (see supplementary fig. S2), along with the higher noise means that the SWD is less sensitive to thin cirrus compared to instruments such as the GOES-16 advanced baseline imager (see supplementary fig. S3). The anvils tracked in this paper have a median retrieved minimum optical depth at $0.64\,\mu m$ of 1.45, although this value is likely biased high as many anvils dissipate at night
when accurate satellite retrievals of optical depth are not available. While this sensitivity captures much of the CRE of DCC anvils (Berry and Mace, 2014) the long lifetimes of dissipating thin anvils may have a significant warming contribution to net anvil CRE (Horner and Gryspeerdt, 2023). As a result, it is expected that the anvil CRE measured in this study is biased low.

An example of the cores and anvils detected by the tobac-flow algorithm is shown in fig. 4, at 3-hour intervals. In fig. 4 a, we see a large number of developing cores over central Africa. In fig. 4 b, we see more developing cores over Western Africa as
the pattern of initiation has shifted with the diurnal cycle. In fig. 4 c,d we observe fewer new developing cores later in the day, but the larger anvil clouds persist into the night-time.

Over the 4-month period of the case study we track a total of 145,463 cores (of which 79,592 are associated with anvil clouds) and 35,941 anvils. Using the detected regions of both core and anvil components of tracked DCCs, the cloud properties and CRE are calculated for each DCC at each time step from the retrieval and broadband fluxes data. The resulting dataset
allows us to analyse the properties of each DCC over their lifetimes from a Lagrangian perspective. While the studied domain contains both land and sea regions, only a small proportion of tracked DCCs occurred over sea (11%), and so we have not separated the analysis of land and oceanic DCCs in this article.

## 4 Results

### 4.1 Spatial and temporal distributions

Figure 5 a shows the frequency of core detections for each 1×1°grid square over the period of the case study. The majority of observed convection occurs over the tropical rainforest regions. During the months of May-August, the Inter-Tropical Convergence Zone (ITCZ) is at its northernmost extent over Africa (Nicholson, 2018). The West African monsoon occurs during these months, with the primary band of convection located between 5-15°N (Nicholson, 2009), which our observations agree with. We observed the maximum frequency of convection at around 6°N, 12°E over the Western High Plateau of Cameroon,
with high frequencies of convection also observed over the Nigerian coastal plains to the West and the Jos Plateau in Northern Nigeria. High rates of convection are also observed over the coastal plains and inland highlands of Guinea, Sierra Leone and Liberia (5–12 °N, 5–15 °W)

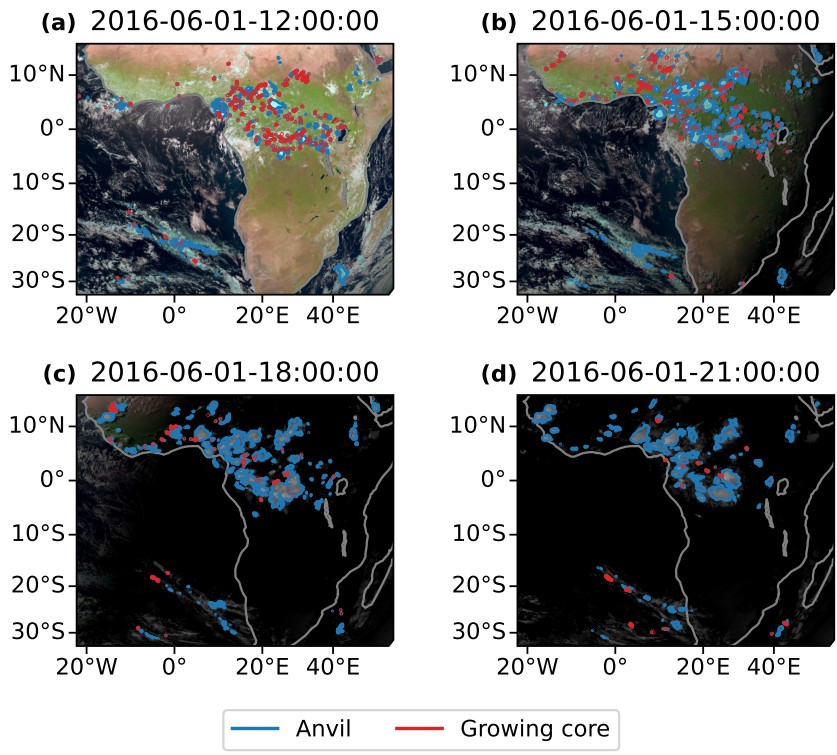

**Figure 4.** An example of the cores (red outline) and anvils (blue outline) detected by tobac-flow plotted over visible composite imagery from SEVIRI, shown at 3-hour time intervals. All times are given in UTC.

Figure 5 b shows the average time of detection for convection in each 1×1°grid square. The average is calculated as the circular mean of the local solar times of core detection in the grid square. Grid squares with a standard deviation greater than 6 hours (indicating a broad spread of initiation times) are given single hatching, and those with standard deviations greater than 12 hours have cross-hatching. The most notable feature of the time of detection is the clear contrast between land and sea. Convection over the land tends to occur in the afternoon (15:00–18:00), whereas over the ocean it occurs between midnight and early morning (00:00–09:00). Furthermore, convection over land tends to occur in a fairly narrow range of times whereas over the ocean convection occurs throughout the diurnal cycle, resulting in the hatching applied to much of the ocean region. There is also a noticeable lake effect on the time of convection occurring over Lake Victoria (2°S, 34°E) and Lake Tanganyika (7°S, 31°E), with convection typically observed in the early morning.

When we compare the regions of Cameroon and Nigeria (4–10°N, 6–14°E), where we detect the most cores in fig. 5 a, with the average time of detection in fig. 5 b, we see that the grid squares with more cores also tend to have an earlier average time of detection than the surrounding grid squares. Precipitation over the Nigerian plains and the Jos Plateau is linked to South-westerly winds bringing moist, warm air from the Gulf of Guinea (Vondou et al., 2010). This warm air may then trigger

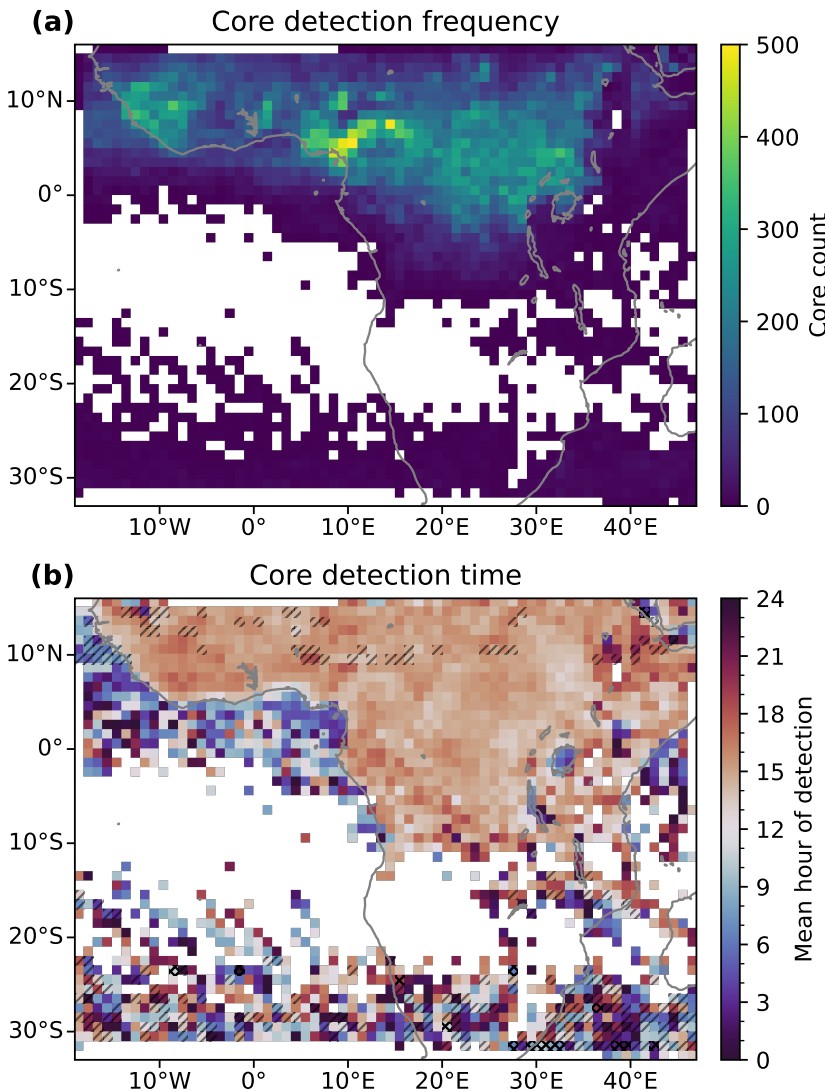

**Figure 5.** a.: The total number of DCC cores detected over the case study for each 1×1°grid box. b.: The average hour of detection for the cores detected in each 1×1°grid box. Grid boxes with a standard deviation greater than 6 hours are single-hatched, and greater than 12 hours cross-hatched.

convection both through the sea breeze effect and orographic lifting when it reaches the highlands, explaining both the higher frequency and earlier timing of convection compared to surrounding regions. A similar relationship between the high frequency of convection and earlier time of detection is also seen over the coastal region and adjacent highlands of Guinea, Sierra Leone and Liberia (5–12°N, 5–15°W) which may be due to the same mechanism.

It should be noted that due to the method of detection, cores that develop under existing anvils are less likely to be detected than those in clear sky regions. As a result, we may underestimate the occurrence of later occurring cores, particularly in regions such as the Northern Sahel where a second, night-time peak of precipitation has been observed.

For all further analysis, we consider only cores and anvils that are detected north of 15°S in order to constrain our analysis to tropical DCCs.

## 4.2    Anvil Cloud Properties

To investigate how the behaviour of DCC anvils is affected by their organisation, we group observed anvils based on how many cores are associated with them, from isolated DCCs with one core to highly-clustered DCCs (such as tropical cloud clusters and MCSs) with 10 or more cores. Anvils with 6–9 cores, and with 10 or more cores, are grouped together to ensure that these groups have a comparable number of members for analysis.

Figure 6 shows properties related to the anvil area and lifetime linked to the number of cores. In fig. 6 a we show the average anvil maximum area for each group. We find that the maximum area increases approximately linearly with the number of cores, with increasingly clustered anvils having increasingly larger maximum areas, and highly clustered anvils having substantially larger anvils. Figure 6 b shows the average anvil lifetime compared to the number of cores. While the lifetime also increases with the number of cores, the difference between isolated and highly clustered anvils is proportionately smaller.

Figure 6 c shows the number of anvils observed with differing numbers of cores. We see that the vast majority of all anvils observed are isolated DCCs, with over 80% having a single detected core. As the number of cores increases, the number of anvils detected decreases rapidly. However, when considering the large increase in both anvil area and lifetime with the number of cores, the total anvil coverage for highly clustered anvils is much larger (see fig. 6 d). Despite their high frequency, isolated DCCs only account for 12% of total anvil coverage, whereas highly clustered (10+ cores) account for over 50%. Previous

studies have found that despite being few in number, MCSs account for the majority of precipitation in Western Africa (Vizy and Cook, 2019).

Figure 7 a shows the average mean CTT, and fig. 7 b the average minimum CTT for anvils with different numbers of cores. While the more clustered anvils have colder average anvil CTT, this decrease plateaus below 220K indicating that the reduction in clear-sky cooling below this temperature may cap the anvil CTT for larger DCCs. The minimum observed CTT within each

anvil, however, are colder and show a greater difference with an increasing number of cores. The most clustered anvils tend to have a minimum CTT of around 180 K, indicating the presence of overshooting tops and the most intense convection. Care should be taken when interpreting such low retrieved CTT values due to the large uncertainty associated with sensor noise at these cold temperatures.

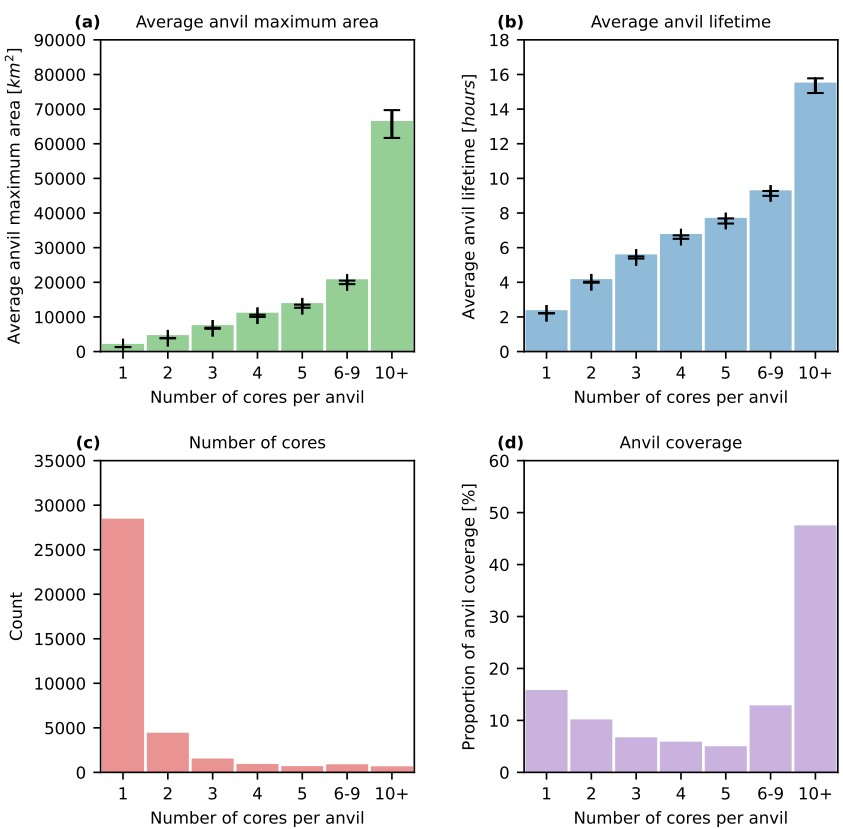

**Figure 6.** Anvil statistics by number of associated cores for a.: average maximum area; b.: average lifetime; c.: the number of observed anvils by number of cores; and d.: percentage of total anvil coverage. Error bars in a and b show the standard error of the mean.

Futyan and Del Genio (2007) divide the DCC lifecycle into growing, mature and dissipating phases based on the time of observation of the coldest anvil CTT, maximum anvil area and dissipation of the anvil. In fig. 8 we show the distribution of the time taken to reach each of these lifecycle milestones for anvils separated by the number of associated cores. For all cases, the average time of minimum anvil CTT occurs before the maximum area, indicating that the anvils continue to grow beyond the maximum of convective activity. As the number of cores associated with each anvil increases, the time of the coldest CTT and largest area occur proportionately earlier during the lifetime of the anvil. As a result, these more clustered anvils spend more of their lifetime existing with warming, shrinking anvils than the isolated DCCs.

In fig. 9, we compare the proportion of the overall anvil lifetime spent in each of the lifecycle phases defined by Futyan and Del Genio (2007) to the number of cores associated with the anvil. There is a clear trend that, as the number of cores increases, the proportion of the lifecycle spent in the growing phase decreases, and the proportion spent in the mature and dissipating phases increases. Although this approach to classifying the lifecycle of anvil clouds is simplistic and does not capture the

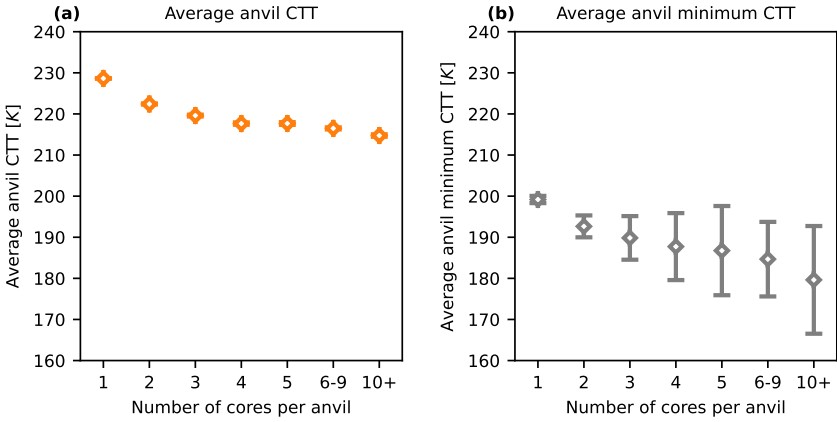

**Figure 7.** Anvil statistics by number of cores for a.: average anvil CTT; and b.: average minimum anvil temperature. Error bars show the standard error of the mean.

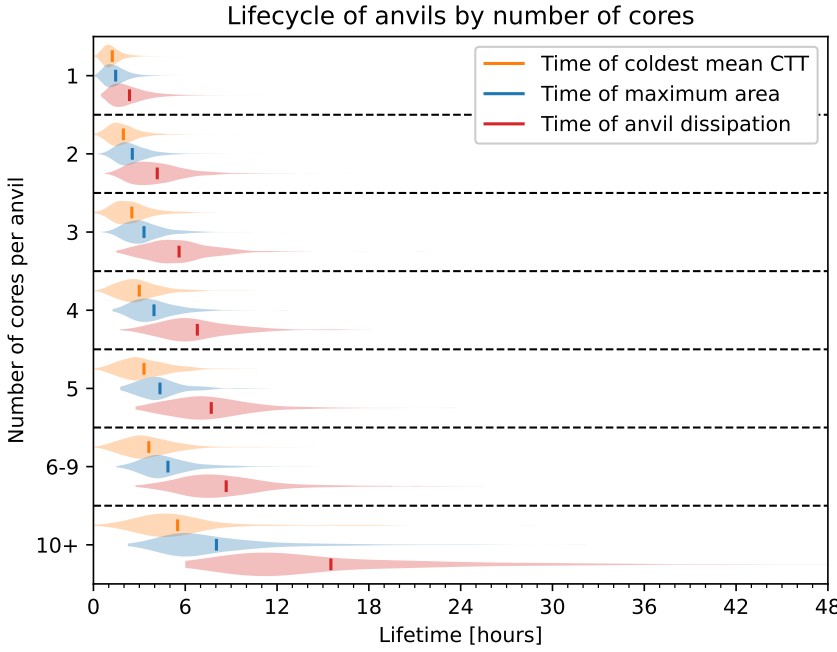

**Figure 8.** The distribution of time to coldest mean anvil CTT (orange), largest anvil area (green) and time of anvil dissipation (red) for anvils grouped by number of cores. The vertical lines show the mean time for each distribution.

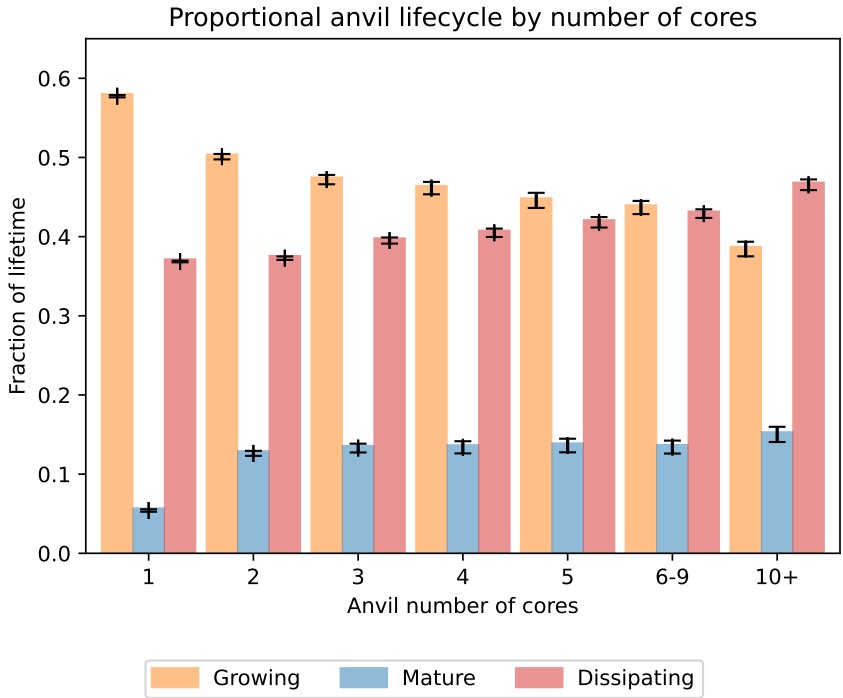

**Figure 9.** The proportion of anvil lifetime spent in the growing (orange), mature (green) and dissipating (red) phase, according to the criteria used by Futyan and Del Genio (2007)

complexities of large, long-lived DCCs which may go through multiple cycles of growth, dissipation and re-invigoration, it can provide a useful perspective when considering the LW CRE of DCCs. The time of the coldest average CTT will be when the LW CRE of the anvil cloud is at its greatest, and so can help understand the evolution of the anvil CRE over its lifetime.

### 4.3 Anvil CRE

Using the broadband fluxes data in conjunction with the tracked DCC dataset, we are able to track how the SW, LW and net CRE evolve over the lifetime of each tracked anvil. Figure 10 shows the time series of SW, LW and net CRE as well as the cumulative average CRE (the average of net CRE over anvil area and lifetime up until that point in time) for several different anvil lifecycles. Note that all fluxes are ToA and measured in the downward direction, so a positive value represents warming and a negative value represents cooling.

Figure 10 a shows the case of an isolated, short-lived DCC. The DCC initiates during the daytime, during which the SW CRE dominates and the net CRE is negative (cooling). However, towards the end of the four-hour lifecycle of the DCC, it transitions to night-time and so while the SW CRE reduces and eventually becomes zero, the LW CRE dominates and the net CRE is positive (warming). While this period of warming moves the cumulative average CRE towards zero, it remains overall

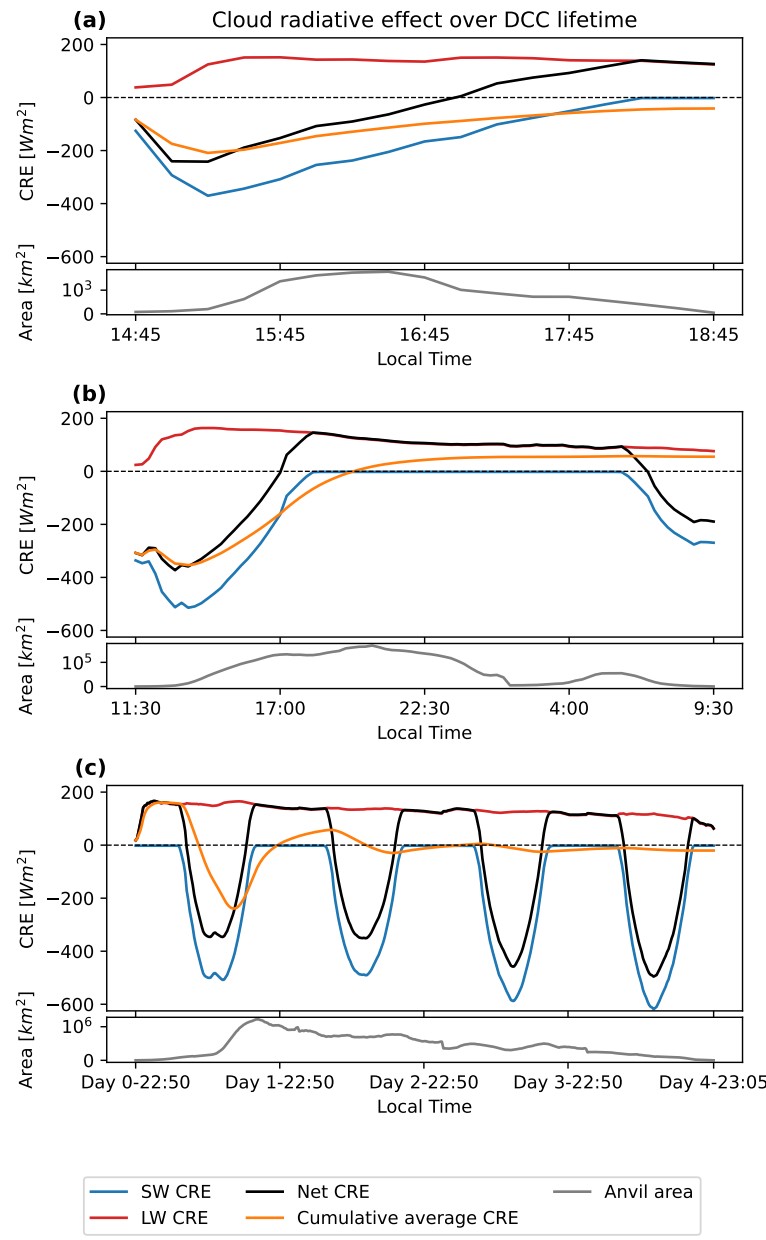

**Figure 10.** Anvil net, LW, and SW CRE, cumulative mean CRE over anvil lifetime and anvil area for a.: an isolated, short-lived (4-hour) DCC; b.: a moderately clustered, 1-day long DCC; and c.: a large, clustered, 4-day long DCC. All times are the local solar time, to the nearest 5-minute interval. The black lines show the change in area of each DCC over their lifecycle.

negative for the overall lifetime of the DCC both due to the longer period spent during the daytime, and the larger area of the anvil cloud during this period.

Figure 10 b shows the case of a longer-lived (22 hours), clustered DCC. It initiates in the morning, and so the SW cooling dominates for the first half of the anvil lifetime. Compared to the isolated DCC, it exists for much longer during the night time, and so the cumulative average becomes positive over the full lifetime of the anvil cloud.

Figure 10 c shows the case of a four-day, highly clustered convective event. In this case, we see the net CRE alternates between warming and cooling throughout the diurnal cycle. The cumulative CRE also alternates between overall warming and cooling throughout the lifetime of the anvil and results in a small net cooling effect.

We see in both the longer-lived cases (fig. 10 b, c) that the LW CRE reduces towards the end of the anvil cloud lifetime. This may be reflective of the findings from fig. 8 that the minimum average CTT occurs before the mid-point of the cloud lifecycle for longer-lived systems. This reduction in LW CRE may be due to a thinning of the anvil cloud (allowing increased LW emission from the surface), or due to heating and stabilisation of the upper troposphere by the DCC. In addition, the cumulative radiative cooling of the anvil top may drive subsidence and reduce the cloud-top height of the anvil over time (Sokol and Hartmann, 2020)

Figure 11 shows the distribution of net lifetime CRE for all tracked anvils. The overall negative average value of $-0.94 \pm 0.91\,\mathrm{Wm^{-2}}$ is very close to zero considering the large spread in CRE. However, the distribution shows a bimodal structure, with two peaks at around $+100\,\mathrm{Wm^{-2}}$ (warming) and $-180\,\mathrm{Wm^{-2}}$ (cooling). The distribution is coloured according to the mean number of cores associated with the anvils in each bin of the distribution. Both the peaks of the distribution are mainly composed of isolated DCCs which occur during the daytime (negative peak) or night-time (positive peak). The centre of the distribution—with average CREs close to zero—shows a greater number of the clustered DCCs with multiple cores which, due to their longer lifetime, tend to exist during both the day- and night time.

In fig. 12 we break down the CRE distribution into that of the SW (fig. 12 a) and LW (fig. 12 b) components. The SW CRE shows a similar bimodal distribution to that of the net CRE, whereas the LW distribution shows a normal distribution. The SW CRE has a large peak at $0\,\mathrm{Wm^{-2}}$ for DCCs that occur during the night-time, and a broad peak centred around $-300\,\mathrm{Wm^{-2}}$ consisting of daytime DCCs, with the average falling between the two. Note that the average for the LW falls to the right of the peak of the distribution because the average is integrated over the anvil area and lifetime, and the largest and longest-lived anvils tend to have colder CTT and hence larger LW CRE.

Figure 13 shows (a) the average instantaneous anvil CRE binned by the time of observation (local solar time) and mean anvil CTT, and (b) the average lifetime anvil CRE binned by time of initial detection (local solar time) and mean anvil CTT. We see that, as expected, mean anvil CRE becomes more positive with decreasing CTT due to increased LW warming. However, the diurnal cycle of detection shows a much stronger contrast, with anvils detected during the daytime having a net cooling effect compared to those at night which have a net warming CRE. This diurnal cycle effect is stronger for those anvils with warmer average CTT, generally representing isolated, shorter-lived DCCs, and is weaker for colder anvil CTT. Note also that in fig. 7 b that the phase of the diurnal cycle shifts to earlier times of detection as average anvil CTT become colder, as these DCCs tend to have longer lifetimes.

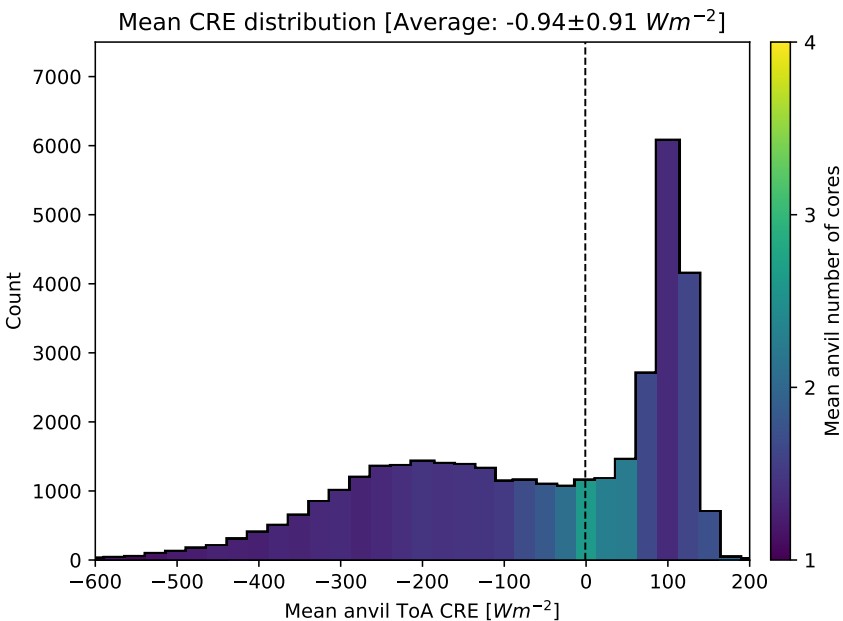

**Figure 11.** The distribution of lifetime anvil CRE for all observed anvils. The mean number of cores per anvil in each bin is indicated by the colour scale. The vertical dashed line shows the integrated mean CRE (over area and lifetime) over all anvils, weighted by the anvil areas ($-0.94 \pm 0.91 \, \mathrm{Wm}^{-2}$).

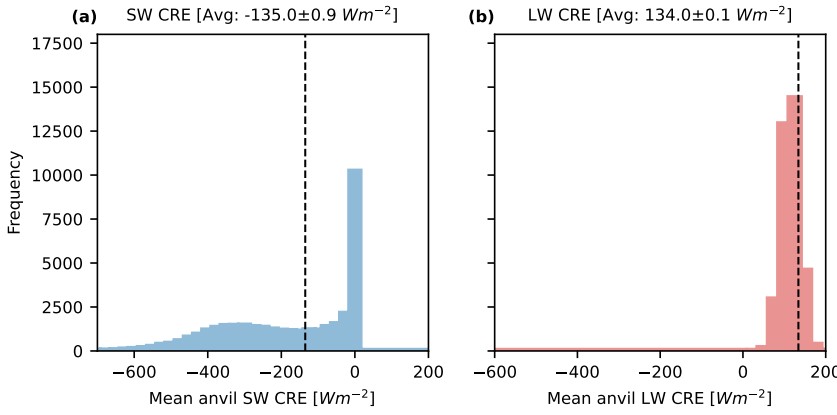

**Figure 12.** The distributions of mean anvil SW CRE (a) and LW CRE (b). The vertical dashed line shows the integrated mean CRE over all anvils (SW: $-135.0 \pm 0.9 \, \mathrm{Wm}^{-2}$, LW: $134.0 \pm 0.1 \, \mathrm{Wm}^{-2}$)

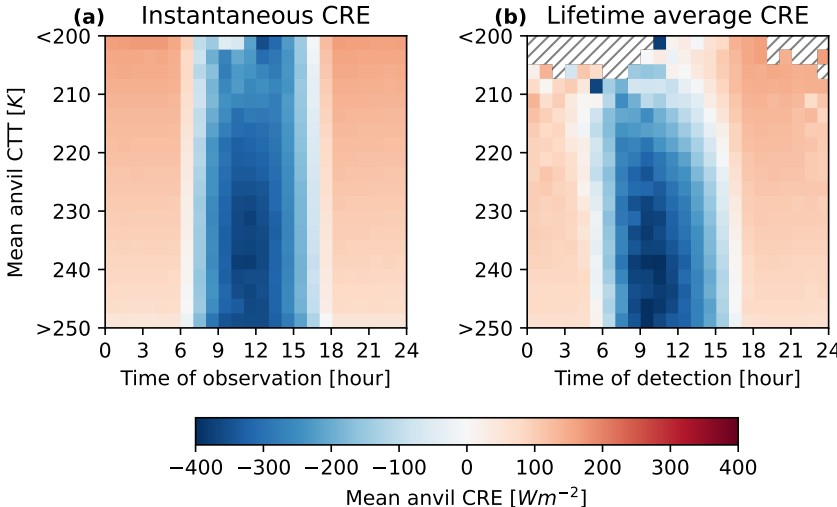

**Figure 13.** (a) Average instantaneous anvil CRE binned by the time of observation (local solar time) and mean anvil CTT. (b) Average lifetime anvil CRE binned by time of initial detection (local solar time) and mean anvil CTT. Hashed regions in (b) show bins in which no anvils were detected.

It is apparent from figs. 11 and 12 that the observed neutral net anvil CRE is not only due to a balance between the SW and LW, but also from a balance of the cooling effect of daytime DCCs and the warming effect of those occurring at night. If the number of DCCs occurring during the daytime were to reduce we would expect a net warming effect without any change to the CRE of individual DCCs. As the diurnal cycle of convection over the ocean is nearly uniform, we should expect little impact on anvil CRE from changes in the time of convective initiation. However, over land, where convective activity is much more common in the afternoon, changes in the diurnal cycle may have a much larger effect on anvil CRE.

Furthermore, fig. 13 b highlights that differences in anvil temperature are linked to the diurnal cycle of anvil CRE as colder anvils tend to have longer lifetimes. As a result, if warming surface temperatures lead to the invigoration of DCCs, the warming effect we would see would be larger than the LW effect from the change in anvil temperature alone. Surface warming may also result in an earlier time of convective initiation, resulting in a cooling feedback.

## 5  Conclusions

By combining a novel cloud tracking algorithm with a new dataset of derived all-sky and clear-sky fluxes from geostationary satellite observations, we were able to detect and track DCC anvils and their associated cores for both isolated and clustered DCCs and investigate their properties, lifecycle and CRE. As this study was performed using data from May-August (Northern

hemisphere summer), we observed the majority of convective activity over the Guinea-Congo rainforest and Savanna regions, as the ITCZ is at its northernmost extent.

We evaluate the degree of convective clustering of each anvil by measuring the number of cores it is associated with. We find that, as expected, anvils with the greatest number of cores—including MCSs—have larger anvil areas, longer lifetimes and the coldest cloud tops. As a result, despite the majority of observed DCCs being isolated, the highly clustered anvils make up most of the anvil coverage, and so cause most of the anvil impact over this region. We also find that the proportion of the lifecycle spent in the mature and dissipating phases increases with the number of cores, and the proportion spent in the growing phase decreases.

When looking into the net CRE of anvils, we find that, although the average CRE across all observed anvils is close to zero, few anvils have near zero CRE themselves. We find a bimodal distribution of anvil CRE, with isolated DCCs that exist during the daytime causing the negative (cooling) peak, and those that exist during the night-time causing the positive (warming) peak. The systems with near zero CRE tend to live longer with more cores, and exist during both the day- and night-time. As a result, when considering the magnitude of the anvil CRE, isolated DCCs have an outsized contribution to the overall average anvil CRE of 21.4% compared to their proportion of all anvil coverage (15.3%) (see supplementary fig. S4).

The interaction between the diurnal cycle of convection and DCC lifetime plays a key role in the shape of the SW anvil CRE distribution and is important to consider in regard to anvil CRE feedback. As the LW CRE is normally distributed, a response to changing cloud top height or temperature may occur as a shift in the distribution. However, the bimodal distribution of the SW CRE must result in more complex adjustments to shift the overall mean. As the position of the peak at $0 \ \mathrm{Wm^{-2}}$ relating to night-time DCCs is fixed, to change the overall average SW CRE either the width of the distribution has to increase or decrease, or the number of DCCs occurring during the day- or night-time has to increase. The former has important implications for the diurnal cycle of temperature in the tropics, and the latter for the diurnal cycle of convection, which, in turn, affects the anvil lifecycle.

Changes in the diurnal cycle of convection may not have a large impact on net anvil CRE over the ocean due to the mostly uniform occurrence of convection throughout the day. Over land, however, the afternoon peak of convection at around 3 pm solar time (see fig. 5) coincides with a time at which anvil CRE is very sensitive to shifts in the diurnal cycle (fig. 13 b). Furthermore, a reduction or increase in the number of DCCs occurring at a specific time of day may change the net CRE of anvils without any change in the CRE of individual DCCs.

Diagnosing a diurnal cycle related anvil cloud feedback in climate models may however be difficult. While Beydoun et al. (2021) found that changes in anvil lifetime contributed little to CRE feedbacks, this study used a radiative-convective-equilibrium model with no diurnal cycle of insolation. Although convective-resolving models have been found to model the diurnal cycle an lifecycle of DCCs better than parameterised climate models (Prein et al., 2015; Feng et al., 2023), but lack good observational constraints. Disentangling the impacts of convective processes and anvil cirrus processes on anvil lifecycle and CRE is also a key challenge. Here, the use of model experiments such as that of Gasparini et al. (2022) may help better understand the impacts of both processes on anvil CRE and the potential for climate feedbacks.

There are, however, a number of limitations in this study which present opportunities for future research. Firstly, as this study only involved 4 months of data during the Northern Hemisphere summer, we were not able to investigate the impact of the seasonal cycle on the behaviour of DCCs and their CRE. Furthermore, extending to a larger domain would allow investigation of regional differences, in particular the important land–sea contrast of deep convection (Takahashi et al., 2023). A major

365    limitation of the SEVIRI data is its poor sensitivity to thin anvil cirrus, which has an important impact on net anvil CRE (Protopapadaki et al., 2017; Horner and Gryspeerdt, 2023). The flexible combined imager (Martin et al., 2021) aboard the third-generation Meteosat may allow better detection and study of thin anvil cirrus over tropical Africa in the near future.

Cloud tracking provides a key capability for the study of deep convective anvil clouds (Gasparini et al., 2023). The ability to observe changes over the lifetime of an anvil cloud independently of changes in the microphysical or macrophysical properties

370    of DCCs. Further application of cloud tracking approaches may better our understanding of DCC lifecycle, its relation to the diurnal cycle of radiation, and its response to a changing climate.

*Code and data availability.*  The CC4CL cloud retrieval algorithm is available for use with the GPL v3.0 licence and can be accessed through the following github repository: https://github.com/ORAC-CC/orac (last accessed: 18 March 2024). The tobac-flow DCC detection and tracking algorithm is available under the BSD 3-clause licence. Version 1.7.6, which was utilised for this study, is archived at the following

375    repository: https://zenodo.org/records/8317062 (Jones, 2023b, last accessed: 18 March 2024). Radiative transfer simulations were performed using libRadtran (Emde et al., 2016). We thank EUMETSAT for the provision of the Meteosat SEVIRI level 1.5 data used in this study, which is openly available via the EUMETSAT data store. The CERES EBAF edition 4.2 data used for calibration were obtained from the NASA Langley Research Center Atmospheric Sciences Data Center.

The dataset of tracked DCCs and their properties used in this study is available at the following repository: https://zenodo.org/records/

380    8317025 (Jones, 2023a, last accessed: 18 March 2024). The material used to prepare this manuscript, including code used to perform analysis and preparation of figures, is archived at the following repository: https://zenodo.org/records/10834939 (Jones, 2024, last accessed: 18 March 2024).

*Author contributions.*  WKJ designed the study. MS produced the dataset of retrieved cloud properties and radiative fluxes. WKJ performed the detection and tracking and the data analysis. WKJ wrote this article with contributions from MS and PS.

385    *Competing interests.*  At least one of the (co-)authors is a member of the editorial board of Atmospheric Chemistry and Physics.

*Acknowledgements.*  The authors acknowledge financial support from the European Research Council (ERC), H2020 European Research Council RECAP (grant no.724602), and from the European Space Agency (ESA) through the Cloud_cci project (contract no.: 4000128637/20/INB). Philip Stier additionally acknowledges funding from the FORCeS and NextGEMs projects under the European Union's Horizon 2020 re-

search program with Grants 821205 and 101003470, respectively. We thank EUMETSAT for providing the Meteosat-11 SEVIRI data used in this study. The processing of retrieved cloud properties and derived broadband fluxes was performed at DWD. The production and analysis of the tracked DCCs dataset were performed on JASMIN: the UK collaborative data analysis facility; and LOTUS: the associated high-performance batch compute cluster.

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
