# Peer review of "A Lagrangian Perspective on the Lifecycle and Cloud Radiative Effect of Deep Convective Clouds Over Africa"

_EGUsphere, 2023_

## Referee Comment (RC1)

Comments on *A Lagrangian perspective on the lifecycle and cloud radiative effect of deep convective clouds over Africa* by Jones, Stengel, & Stier

This paper by Jones et al. presents a Lagrangian analysis of anvil cloud properties over West Africa. By focusing on anvil clouds over land, the authors address an important gap in the anvil-radiation-climate literature, which in the past has mostly focused on maritime cloud systems. The synthesis of radiometer measurements, cloud property retrievals, radiative flux estimates, and a complex cloud-tracking algorithm is an impressive technical feat and makes a valuable contribution to the literature.

The text is well written and the figures are clear. The authors' technical expertise in cloud tracking and deep familiarity with the anvil cloud literature is also clear. Overall, the paper was a pleasure to read and I learned a lot from it.

My comments are mostly questions or relatively minor points of clarification. They can be addressed with some additional text and do not require further analysis, although there a few optional suggestions for small additions that could aid interpretation. For this reason, my recommendation is that the paper be accepted pending minor revisions.

Signed,

Adam Sokol
University of Washington, Seattle

**Comments/Points of clarification**

1. The authors importantly note that SEVIRI cannot reliably detect thin anvil, which have an important impact on net anvil CRE. Observational analyses over tropical oceans show that thin anvil cirrus with optical depth 1-2 are much more abundant than thicker anvils and, on average, have the largest effect on the TOA CRE (e.g., Berry & Mace 2014 Fig 12b; Hartmann & Berry 2017; Sokol & Hartmann 2020 Fig 1). It would be useful to provide an approximate optical depth (or similar metric) threshold that separates the anvils that are included in the analysis from those that are not. A threshold near $\tau \sim 1$ has very different implications than one near $\tau \sim 4$, etc.
   This limitation means that the mean CRE results are almost certainly biased low. While my hunch is that thin anvils are less abundant over land than in the maritime regions examined in those previous studies, I would not be surprised if the bias is of comparable magnitude to, or even larger than, the CERES bias correction. Since quantifying this bias can't be done easily, I think it would be fine just to note an approximate optical depth threshold and its implications for the results (i.e., that CRE is biased low, if that is something the authors agree with).

2. The authors might take interest in Gasparini et al 2022 (https://doi.org/10.1175/JCLI-D-21-0211.1), a modeling study that addresses some of the same questions addressed in this paper. Some of their results (e.g., their Figs 3 & 4) are quite relevant here and could serve as an important and useful point of comparison at several points throughout the paper. Lines 41-43 is where this paper first came to mind.

3. Is the CRE bias correction applied uniformly as the average over the entire region, or is it applied point-by-point using the spatially varying biases shown in Fig 3?

4. After reading through section 3 a few times, I was still unsure how the Lagrangian tracking system described on lines 148-162 differs from the two types of tracking systems mentioned earlier in the section. How does it overcome the issue described on line 145 (that the motion of a large system cannot be accurately described as a single vector)? I may be misunderstanding something here, but it could be worth adding a sentence that more clearly highlights the improvements that are being made over previous algorithms.

5. It would be useful to have a more detailed description of how the number of cores associated with a particular anvil is determined. This variable is used to sort the data in many of the figures. Does the number of cores reflect the total number of connected cores over the whole anvil life cycle, or do the cores have to appear simultaneously? How is an anvil associated with multiple cores if cores developing under existing anvils cannot be detected? Does this mean that the cores need to develop in close proximity at the same moment in time? I imagine many anvils might initially develop from a single convective core but merge with other anvils later on. It could be nice to have some more detail on this, even if it is only included as a supplement.

6. Is the analysis limited to anvils over land, or are all anvils included? Fig 5 shows that land is dominating the statistics either way, but I was unsure of this.

7. My interpretation of Fig 13 is that it shows the instantaneous CRE of anvil clouds as a function of CTT and time of day. But the sentence on lines 282-283 beginning with "Note", as well as the following paragraph, are talking about the lifetime-averaged CRE of anvils. This was a bit confusing. It is suggested on 282-283 that the CTT-dependence of the diurnal cycle of CRE is due to the different cloud lifetimes associated with different CTT. But if Fig 13 shows instantaneous CRE, it is hard to make any conclusions that depend on cloud lifetime.
   It could be a nice addition to show a figure that is similar to Fig 13 but showing the lifetime-integrated CRE as a function of CTT and the *initiation* time of the convective system. This would make it easy to interpret the combined effects of CTT and lifetime on CRE.

**Minor line comments**

8. Line 62-63: I think the arguments of Seeley et al (2019) are slightly misrepresented here. They found that the radiative tropopause temperature (which differs from the "inversion temperature", i.e. the cold point) was fixed across a wide range of surface temps. But they do not attribute this to FAT physics, which they argue are a weak constraint.

9. Line 69: there is a very recent observational analysis by Liu et al (2023) finding a decreasing trend in CTT that the authors may wish to include here. Reference below.

10. Line 83-84: typo "…to investigate both the CRE of individual CREs, as well as…"

11. Line 137-138: "Firstly, those…" not a complete sentence

12. Line 140-141: "Secondly, those…" not a complete sentence

13. Line 244: "CREover"

14. Line 265: "The overall negative average value of -8.17 W/m2 is approximately zero when considering the negative bias…". Earlier it is mentioned that the mean bias is -3.67 W/m2, which would bring the

total to -4.5 +/- 0.85 W/m2, which is different from zero. I am probably misinterpreting how the bias is applied (see comment #3 above).

15. Line 279: I think this should be "mean anvil CRE becomes more positive with **decreasing** CTT"

16. Line 280-281: I think this should be "This diurnal cycle effect is stronger for those anvils with **warmer** CTT"

**References from above that are not already in the paper:**

Berry, E., & Mace, G. G. (2014). Cloud properties and radiative effects of the Asian summer monsoon derived from A-Train data. Journal of Geophysical Research, 119(15), 9492–9508. https://doi.org/10.1002/2014JD021458

Gasparini, B., Sokol, A. B., Wall, C. J., Hartmann, D. L., & Blossey, P. N. (2022). Diurnal Differences in Tropical Maritime Anvil Cloud Evolution. Journal of Climate, 35(5), 1655–1677. https://doi.org/10.1175/JCLI-D-21-0211.1

Hartmann, D. L., & Berry, S. E. (2017). The balanced radiative effect of tropical anvil clouds. Journal of Geophysical Research, 122(9), 5003–5020. https://doi.org/10.1002/2017JD026460

Liu, H., Koren, I., & Altaratz, O. (2023). Observed decreasing trend in the upper-tropospheric cloud top temperature. Npj Climate and Atmospheric Science, 6(1), Article 1. https://doi.org/10.1038/s41612-023-00465-5

---

## Referee Comment (RC2)

Review of "A Lagrangian Perspective on the Lifecycle and Cloud Radiative Effect of Deep Convective Clouds Over Africa" by W. K. Jones et al.

Jones et al. analyze geostationary satellite observations to investigate the diurnal cycle and radiative effects of tropical deep convective clouds over Africa and the tropical Atlantic Ocean. They use a novel cloud-tracking algorithm that allows them to investigate the clouds from a Lagrangian point of view. This analysis shows that individual anvil clouds can have a wide range of radiative effects depending on the time of day that they initiate. Thus, changes in the diurnal cycle of convective cloud be an important and underappreciated climate-feedback mechanism.

I believe that the research topic is highly relevant, the analysis is well done, and the writing and figures are clear and concise. I have only a few comments and suggestions for improvements, which are listed below. I therefore recommend *minor revision* for the manuscript.

**General Comments**

My only main comment about the paper is that the discussion about how the results relate to the existing cloud-climate feedback literature is not as specific as I hoped it would be. The authors make a compelling case that changes in the diurnal cycle of convection could be an important and understudied climate-feedback mechanism, but the discussion about how to address this challenge is not very clear. Can the results of the current study help to estimate the diurnal-cycle-induced climate feedback? If not, then what are some ways that we might make progress on this in the future? Have any physical mechanisms been proposed that would change the timing or amplitude of the convective diurnal cycle as the climate changes? Does the community have the necessary analysis methods to diagnose this feedback? As far as I know, none of the current methods of cloud-feedback analysis can diagnose feedbacks from changes in the diurnal cycle of clouds, so I'm not even sure that the community has the proper tools to study this rigorously. I think that a more specific discussion about how the results relate to the existing cloud-feedback literature and potential future directions would improve the end of the paper. It would also align well with the introduction, which discusses anvil-cloud feedback mechanisms at length.

**Specific Comments**

Line 161 "we only detect and track the thick portion of the anvil in this article": Can you be more specific about what "thick portion" means? For example, can you state the minimum cloud visible optical thickness that can be tracked by the algorithm?

Section 4.2: I think the current analysis in this section is well done, but I wonder if an even stronger signal would emerge if the analysis was performed separately with land-based convection and ocean-based convection. I think that oceanic clouds are typically larger, longer lasting, and have less intense convection than land-based clouds, so the land-ocean contrast may alias into the statistics in Fig. 6 and Fig. 7.

Line 284: This paragraph is written in a way that seems to imply that the average anvil-cloud net CRE must remain near zero as the climate changes. I'm not aware of any

convinced physical mechanism or conservation law that would require the net anvil-cloud CRE to remain near zero. Can you please explain why you think it will remain near zero or acknowledge the possibility that it will not remain near zero?

**Technical Corrections**
Line 15: The word "distribution" is used twice in the sentence. Consider changing to "We find that the anvil cloud CRE of our tracked DCCs has a bimodal distribution."

Line 227 (and elsewhere): I think the name "Genio" should be "Del Genio"

Line 279 "We see that, as expected, mean anvil CRE becomes more positive with increasing CTT": Should this be "mean anvil CRE becomes less positive or more negative …"

Line 308: change "outsize" to "outsized"

---

## Author Comment (AC1)

Reviewer comments in italics
Author comments in upright text

Comments on *A Lagrangian perspective on the lifecycle and cloud radiative effect of deep convective clouds over Africa by Jones, Stengel, & Stier*

*This paper by Jones et al. presents a Lagrangian analysis of anvil cloud properties over West Africa. By focusing on anvil clouds over land, the authors address an important gap in the anvil-radiation-climate literature, which in the past has mostly focused on maritime cloud systems. The synthesis of radiometer measurements, cloud property retrievals, radiative flux estimates, and a complex cloud-tracking algorithm is an impressive technical feat and makes a valuable contribution to the literature.*

*The text is well written and the figures are clear. The authors' technical expertise in cloud tracking and deep familiarity with the anvil cloud literature is also clear. Overall, the paper was a pleasure to read and I learned a lot from it.*

*My comments are mostly questions or relatively minor points of clarification. They can be addressed with some additional text and do not require further analysis, although there a few optional suggestions for small additions that could aid interpretation. For this reason, my recommendation is that the paper be accepted pending minor revisions.*

*Signed,*

*Adam Sokol*
*University of Washington, Seattle*

**Comments/Points of clarification**

1. *The authors importantly note that SEVIRI cannot reliably detect thin anvil, which have an important impact on net anvil CRE. Observational analyses over tropical oceans show that thin anvil cirrus with optical depth 1-2 are much more abundant than thicker anvils and, on average, have the largest effect on the TOA CRE (e.g., Berry & Mace 2014 Fig 12b; Hartmann & Berry 2017; Sokol & Hartmann 2020 Fig 1). It would be useful to provide an approximate optical depth (or similar metric) threshold that separates the anvils that are included in the analysis from those that are not. A threshold near $\tau$~1 has very different implications than one near $\tau$~4, etc.*
      *This limitation means that the mean CRE results are almost certainly biased low. While my hunch is that thin anvils are less abundant over land than in the maritime regions examined in those previous studies, I would not be surprised if the bias is of comparable magnitude to, or even larger than, the CERES bias correction. Since quantifying this bias can't be done easily, I think it would be fine just to note an approximate optical depth threshold and its implications for the results (i.e., that CRE is biased low, if that is something the authors agree with).*

We agree that this is an important point, and have conducted a number of idealized, 1D radiative transfer simulations using libRadtran to assess the sensitivity of SEVIRI to anvil clouds at different heights and optical depths. We find that, using the detection thresholds in this study, the thick anvil detection is sensitive to optical depths between 1 and 1.5. This is backed up by the median of the minimum retrieved optical depth of tracked anvils of 1.45, and this value is likely high due to the inability to accurately retrieve OD at nighttime when many anvils dissipate. While this captures most of the CRE of anvil clouds, we agree that all CRE values are likely biased low by the inability to detect thin anvils. The following paragraph has been updated:

"Due to the lack of sensitivity of the SEVIRI SWD to thin ice clouds, we only detect and track the thick portion of the anvil in this article. The WVD channel of SEVIRI is capable of detecting anvils with optical thicknesses of approximately 1–1.5 (see supplementary fig. S1). However, the closer spacing and narrower bandwidth of the SEVIRI LW window channels (see supplementary fig. S2), along with the higher noise means that the SWD is less sensitive to thin cirrus compared to instruments such as the GOES-16 ABI (see supplementary fig. S3). The anvils tracked in this paper have a median retrieved minimum optical depth of 1.45, although this value is likely biased high as many anvils dissipate at night when accurate satellite retrievals of optical depth are not available. While this sensitivity captures much of the CRE of DCC anvils (Berry and Mace, 2014) the long lifetimes of dissipating thin anvils may have a significant warming contribution to net anvil CRE (Horner and Gryspeerdt, 2023). As a result, it is expected that the anvil CRE measured in this study are biased low."

In addition, a number of supplementary figures have been added, which are reproduced below:

[Figure]

**Figure S1.** Simulated sensitivity of the SEVIRI 10.8 μm BT (top) and WVD minus SWD (bottom) to anvil clouds of varying optical thickness at heights of 10, 12 and 14 km. The LibRadTran model was used to estimate the observed radiances, and all simulations used ice clouds with cloud top particle effective radius of 20 μm. The grey dashed line shows the 241 K BT, which, although commonly used as a threshold for anvil detection in satellite imagery, shows large sensitivity of the minimum optical thickness detected with the height of the anvil cloud. The grey region in the lower plot shows the range of temperatures in which the edge of the anvil is detected, as described in Jones et al. (2023). Similar sensitivity is found for all three cloud heights, with the optical depths of around 1–1.5 seen in the middle of the hysteresis region. The median minimum retrieved optical depth of all tracked anvils in our dataset is found to be 1.45, although this value is biased high by the inability to retrieve optical depth accurately at night-time.

[Figure]

**Figure S2.** Comparison of the relative spectral response (RSR) functions for the GOES-16 ABI and Meteosat-11 SEVIRI thermal IR channels. The LW window channels on ABI (channels 13 and 15) have a wider spacing than those of SEVIRI (channels IR10.8 and IR12.0). This wider spacing allows ABI to be more sensitive to the emissivity difference of ice clouds at wavelengths between 10 and 12 mum, and so it is better able to detect thin cirrus clouds.

[Figure]

**Figure S3.** Comparison of the sensitivities of ABI (dashed lines) and SEVIRI (solid lines) to anvil clouds of different optical thickness, using the LibRadTran simulation of an anvil at 14 km as used in fig. S1. The 10.8 μm BT (top panel) and WVD (middle panel) show very similar values for both instruments. The simulations of the SWD (bottom panel) show that SEVIRI is only about half as sensitive as ABI to thin ice clouds.

2. *The authors might take interest in Gasparini et al 2022 ([https://doi.org/10.1175/JCLI-D-21-0211.1](https://doi.org/10.1175/JCLI-D-21-0211.1)), a modeling study that addresses some of the same questions addressed in this paper. Some of their results (e.g., their Figs 3 & 4) are quite relevant here and could serve as an important and useful point of comparison at several points throughout the paper. Lines 41-43 is where this paper first came to mind.*

Gasparini et al 2022 provides great insight into how the diurnal cycle impacts anvil cloud evolution. We have included it in discussed of how such an approach could be used to separate the diurnal effects of convective processes and cirrus processes when investigating anvil cloud properties in the conclusion. See below:

"Diagnosing a diurnal cycle related anvil cloud feedback in climate models may however be difficult. Beydoun et al. (2021) found that changes in anvil lifetime contributed little to CRE feedbacks in a cloud-resolving radiative-convective equilibrium model. It is unclear how well the diurnal cycle of convection and convective lifecycle are represented in such a model, although convective-resolving models have been found to model these better than parameterised climate models (Prein et al., 2015; Feng et al., 2023). Disentangling the impacts of convective processes and anvil cirrus processes on anvil lifecycle and CRE is also a key challenge. Here, the use of model experiments such as that of Gasparini et al. (2022) may help better understand the impacts of both processes on anvil Cloud Radiative Effect (CRE) and the potential for climate feedbacks."

3. *Is the CRE bias correction applied uniformly as the average over the entire region, or is it applied point-by-point using the spatially varying biases shown in Fig 3?*

Bias correction has been applied uniformly over the entire region. Line 129 has been updated to the following to make this clear:

"Correction for these biases have been applied uniformly to all further CRE values given in this article."

4. *After reading through section 3 a few times, I was still unsure how the Lagrangian tracking system described on lines 148-162 differs from the two types of tracking systems mentioned earlier in the section. How does it overcome the issue described on line 145 (that the motion of a large system cannot be accurately described as a single vector)? I may be misunderstanding something here, but it could be worth adding a sentence that more clearly highlights the improvements that are being made over previous algorithms.*

The Lagrangian approach calculates a motion vector for each pixel in the satellite imagery, rather than s single motion vector for each detected DCC. As a result, complex motions such as divergence, convergence and rotation can be better handled during detection and tracking. The paragraph beginning at L148 has been updated to include this information:

"To approach the challenge of tracking both isolated DCCs and large, clustered systems, we address the role of cloud motion in the scaling problem. tobac-flow first estimates the motion of DCCs at each pixel using an optical-flow algorithm. Then, using these estimated motion vectors, we construct a semi-Lagrangian framework in which to perform the detection and tracking. This approach addresses two issues found in traditional cloud tracking approaches. First, estimating a motion vector for each pixel allows complex motions (including divergence, rotation, splitting and merging) to be compensated for, rather than just the bulk motion found using the centroid tracking methods. Second, by estimating the cloud motion a priori, we are able to use this information within the detection step of the algorithm, and can separate changes in cloud properties such as BT from those observed due to cloud motion. This framework removes the problem of DCC motion, allowing us to track both isolated and large DCCs at the same time."

5. *It would be useful to have a more detailed description of how the number of cores associated with a particular anvil is determined. This variable is used to sort the data in many of the figures. Does the number of cores reflect the total number of connected cores over the whole anvil life cycle, or do the cores have to appear simultaneously? How is an anvil associated with multiple cores if cores developing under existing anvils cannot be detected? Does this mean that the cores need to develop in close proximity at the same moment in time? I imagine many anvils might initially develop from a single convective core but merge with other anvils later on. It could be nice to have some more detail on this, even if it is only included as a supplement.*

And anvil and core are associated if the two overlap while the core is in the upper troposphere. The total number of cores per anvil counts all cores associated with an anvil over its lifetime, with no requirement for close proximity or timing and including all merges and splits. We often see the case as you describe that a large MCSs begins as a single core or several merging isolated systems. We have added the following details to explain this:

"Starting from these convective cores, we then detect the surrounding anvil cloud using the WVD field (Müller et al., 2018, 2019) and continue to detect the anvil until its dissipation, even after the core is no longer visible. A core and anvil are linked with each other if the two overlap at a time when the core has mean WVD of > 5 K, indicating that it has reached the upper troposphere. Each anvil cloud can be associated with multiple cores, allowing us to identify cases of clustered convection. As we detect the cores based on cloud-top cooling, however, we can only detect the cores themselves during the growing phase, and cannot detect cores that occur underneath cold, high, anvil clouds. When determining the number of cores associated with an anvil, we count the total number of cores observed over the entire lifetime, even if they do not occur at the same time, and including all merges and splits of the anvil cloud. This flexibility allows for a wide range of different organised systems to be analysed."

6. *Is the analysis limited to anvils over land, or are all anvils included? Fig 5 shows that land is dominating the statistics either way, but I was unsure of this.*

The analysis includes both anvils over land and sea, although as the domain area covers mostly land only a small proportion of DCCs (11%) are detected occurring over the sea. We have added the following sentence to make this clear:
      "While the studied domain contains both land and sea regions, only a small proportion of tracked DCCs occurred over sea (11%), and so we have not separated the analysis of land and oceanic DCCs in this article."

7. *My interpretation of Fig 13 is that it shows the instantaneous CRE of anvil clouds as a function of CTT and time of day. But the sentence on lines 282-283 beginning with "Note", as well as the following paragraph, are talking about the lifetime-averaged CRE of anvils. This was a bit confusing. It is suggested on 282-283 that the CTT-dependence of the diurnal cycle of CRE is due to the different cloud lifetimes associated with different CTT. But if Fig 13 shows instantaneous CRE, it is hard to make any conclusions that depend on cloud lifetime.*
    *It could be a nice addition to show a figure that is similar to Fig 13 but showing the lifetime-integrated CRE as a function of CTT and the initiation time of the convective system. This would make it easy to interpret the combined effects of CTT and lifetime on CRE.*

Figure 13 shows the lifetime CRE of anvils, however the comparison between instantaneous and lifetime CRE

is interesting and so we have updated it to include both cases. The paragraph beginning L278 has been updated to the following:

"Figure 13 shows (a) the average instantaneous anvil CRE binned by the time of observation (local solar time) and mean anvil CTT, and (b) the average lifetime anvil CRE binned by time of initial detection (local solar time) and mean anvil CTT. We see that, as expected, mean anvil CRE becomes more positive with decreasing CTT due to increased LW warming. However, the diurnal cycle of detection shows a much stronger contrast, with anvils detected during the daytime having a net cooling effect compared to those at night which have a net warming CRE. This diurnal cycle effect is stronger for those anvils with warmer average CTT, generally representing isolated, shorter-lived DCCs, and is weaker for colder anvil CTT. Note also that in fig. 7 b that the phase of the diurnal cycle shifts to earlier times of detection as average anvil CTT become colder, as these DCCs tend to have longer lifetimes."

Figure 13 has been updated to the following:

[Figure]

**Figure 13.** (a) Average instantaneous anvil CRE binned by the time of observation (local solar time) and mean anvil CTT. (b) Average lifetime anvil CRE binned by time of initial detection (local solar time) and mean anvil CTT. Hashed regions in (b) show bins in which no anvils were detected.

**_Minor line comments_**

8. _Line 62-63: I think the arguments of Seeley et al (2019) are slightly misrepresented here. They found that the radiative tropopause temperature (which differs from the "inversion temperature", i.e. the cold point) was fixed across a wide range of surface temps. But they do not attribute this to FAT physics, which they argue are a weak constraint._

Yes, the statement was clumsily written. It has now been updated and expanded to the following to better represent their arguments:
"Seeley et al. (2019), argued that FAT is a weak constraint on anvil temperature as while the radiative tropopause temperature remains fixed, the temperature of the tropopause lapse rate inversion can vary widely. Furthermore, as anvils tend to detrain below the tropopause, (Takahashi et al., 2017; Wang et al., 2020), anvil temperature and the tropopause temperature may only be weakly connected. Seidel and Yang (2022) however found the inclusion of $CO_2$ radiative heating produces anvil temperatures consistent with FAT."

9. _Line 69: there is a very recent observational analysis by Liu et al (2023) finding a decreasing trend in CTT that the authors may wish to include here. Reference below._

Yes, this helps show how convective processes may influence anvil CRE. A new sentence has been included at the end of this paragraph:

"Multi-decadal satellite observations have shown a cooling of upper tropospheric cloud temperature over land (Liu et al., 2023), indicating that changes in convective processes may lead to stronger cooling feedbacks"

*10. Line 83-84: typo "…to investigate both the CRE of individual CREs, as well as…"*

Fixed to "…to investigate both the CRE of individual DCCs, as well as…"

*11. Line 137-138: "Firstly, those…" not a complete sentence*

*12. Line 140-141: "Secondly, those…" not a complete sentence*

11 & 12 have been reworded. The paragraph now reads as follows:

"Algorithms for the detection and tracking of DCCs in satellite imagery have generally been developed for one of two applications: tracking convective cores and isolated convection, or tracking large MCS anvils. Those designed for tracking deep convective cores, or isolated DCCs, include Cb-TRAM (Zinner et al., 2008, 2013) or tobac (Heikenfeld et al., 2019). These algorithms work by detecting regions of convective updraft or a proxy (such as cloud top cooling rate), and then treating these regions as point-like objects that are advected over time. The second group, designed for tracking MCSs, include algorithms such as PyFLEXTRKR (Feng et al., 2022), TAMS (Núñez Ocasio et al., 2020) or TOOCAN (Fiolleau and Roca, 2013). These algorithms detect large regions of cold cloud tops which indicate anvil clouds, and then link them over time by overlapping regions at subsequent time steps."

*13. Line 244: "CREover"*

Corrected

*14. Line 265: "The overall negative average value of -8.17 W/m2 is approximately zero when considering the negative bias…". Earlier it is mentioned that the mean bias is -3.67 W/m2, which would bring the total to -4.5 +/- 0.85 W/m2, which is different from zero. I am probably misinterpreting how the bias is applied (see comment #3 above).*

This sentenced was mistakenly left over from an early draft of the article. It has now been corrected to: "The overall negative average value of $-0.94 \pm 0.91$ Wm$^{-2}$ is very close to zero considering the large spread in CRE"

*15. Line 279: I think this should be "mean anvil CRE becomes more positive with **decreasing** CTT"*

*16. Line 280-281: I think this should be "This diurnal cycle effect is stronger for those anvils with **warmer** CTT"*

15 & 16 Corrected

***References from above that are not already in the paper:***

*Berry, E., & Mace, G. G. (2014). Cloud properties and radiative effects of the Asian summer monsoon derived from A-Train data. Journal of Geophysical Research, 119(15), 9492–9508.*

*https://doi.org/10.1002/2014JD021458*

*Gasparini, B., Sokol, A. B., Wall, C. J., Hartmann, D. L., & Blossey, P. N. (2022). Diurnal Differences in Tropical Maritime Anvil Cloud Evolution. Journal of Climate, 35(5), 1655–1677. https://doi.org/10.1175/JCLI-D-21-0211.1*

*Hartmann, D. L., & Berry, S. E. (2017). The balanced radiative effect of tropical anvil clouds. Journal of Geophysical Research, 122(9), 5003–5020. https://doi.org/10.1002/2017JD026460*

*Liu, H., Koren, I., & Altaratz, O. (2023). Observed decreasing trend in the upper-tropospheric cloud top temperature. Npj Climate and Atmospheric Science, 6(1), Article 1. https://doi.org/10.1038/s41612-023-00465-5*

Additional references:

Berry, E. and Mace, G. G.: Cloud Properties and Radiative Effects of the Asian Summer Monsoon Derived from A-Train Data, Journal of Geophysical Research: Atmospheres, 119, 9492–9508, https://doi.org/10.1002/2014JD021458, 2014.

Horner, G. and Gryspeerdt, E.: The Evolution of Deep Convective Systems and Their Associated Cirrus Outflows, Atmospheric Chemistry and Physics, 23, 14 239–14 253, https://doi.org/10.5194/acp-23-14239-2023, 2023

Gasparini, B., Sokol, A. B., Wall, C. J., Hartmann, D. L., and Blossey, P. N.: Diurnal Differences in Tropical Maritime Anvil Cloud Evolution, Journal of Climate, 35, 1655–1677, https://doi.org/10.1175/JCLI-D-21-0211.1, 2022.

Prein, A. F., Langhans, W., Fosser, G., Ferrone, A., Ban, N., Goergen, K., Keller, M., Tölle, M., Gutjahr, O., Feser, F., Brisson, E., Kollet, S., Schmidli, J., van Lipzig, N. P. M., and Leung, R.: A Review on Regional Convection-Permitting Climate Modeling: Demonstrations, Prospects, and Challenges, Reviews of Geophysics, 53, 323–361, https://doi.org/10.1002/2014RG000475, 2015.

Feng, Z., Leung, L. R., Hardin, J., Terai, C. R., Song, F., and Caldwell, P.: Mesoscale Convective Systems in DYAMOND Global Convection-Permitting Simulations, Geophysical Research Letters, 50, e2022GL102 603, https://doi.org/10.1029/2022GL102603, 2023

Beydoun, H., Caldwell, P. M., Hannah, W. M., and Donahue, A. S.: Dissecting Anvil Cloud Response to Sea Surface Warming, Geophysical Research Letters, 48, e2021GL094 049, https://doi.org/10.1029/2021GL094049, 2021

Seidel, S. D. and Yang, D.: Temperatures of Anvil Clouds and Radiative Tropopause in a Wide Array of Cloud-Resolving Simulations, Journal of Climate, 35, 8065–8078, https://doi.org/10.1175/JCLI-D-21-0962.1, 2022.

---

## Author Comment (AC2)

Reviewer comments in italics

Author comments in upright text

*Review of "A Lagrangian Perspective on the Lifecycle and Cloud Radiative Effect of Deep Convective Clouds Over Africa" by W. K. Jones et al.*

*Jones et al. analyze geostationary satellite observations to investigate the diurnal cycle and radiative effects of tropical deep convective clouds over Africa and the tropical Atlantic Ocean. They use a novel cloud-tracking algorithm that allows them to investigate the clouds from a Lagrangian point of view. This analysis shows that individual anvil clouds can have a wide range of radiative effects depending on the time of day that they initiate. Thus, changes in the diurnal cycle of convective cloud be an important and underappreciated climate-feedback mechanism.*

*I believe that the research topic is highly relevant, the analysis is well done, and the writing and figures are clear and concise. I have only a few comments and suggestions for improvements, which are listed below. I therefore recommend minor revision for the manuscript.*

**General Comments**

*My only main comment about the paper is that the discussion about how the results relate to the existing cloud-climate feedback literature is not as specific as I hoped it would be. The authors make a compelling case that changes in the diurnal cycle of convection could be an important and understudied climate-feedback mechanism, but the discussion about how to address this challenge is not very clear. Can the results of the current study help to estimate the diurnal-cycle-induced climate feedback? If not, then what are some ways that we might make progress on this in the future? Have any physical mechanisms been proposed that would change the timing or amplitude of the convective diurnal cycle as the climate changes? Does the community have the necessary analysis methods to diagnose this feedback? As far as I know, none of the current methods of cloud-feedback analysis can diagnose feedbacks from changes in the diurnal cycle of clouds, so I'm not even sure that the community has the proper tools to study this rigorously. I think that a more specific discussion about how the results relate to the existing cloud-feedback literature and potential future directions would improve the end of the paper. It would also align well with the introduction, which discusses anvil-cloud feedback mechanisms at length.*

We have included further discussion on this topic in the conclusion. It is, however, still uncertain whether we can investigate such a feedback at present. The traditional approach to assessing anvil feedbacks using GCMs is unlikely to be appropriate as they do a poor job representing the diurnal cycle and lifecycle of DCCs. While convective-resolving models are better in this regard they are not yet well constrained by observations. In addition, separating the effects of convective processes and cirrus processes on anvils is a major challenge. We have included the two paragraphs below discussing this:

"Changes in the diurnal cycle of convection may not have a large impact on net anvil CRE over the ocean due to the mostly uniform occurrence of convection throughout the day. Over land, however, the afternoon peak of convection at around 3 pm solar time (see fig. 5) coincides with a time at which anvil CRE is very sensitive to shifts in the diurnal cycle (fig. 13 b). Furthermore, a reduction or increase in the number of DCCs occurring at a specific time of day may change the net CRE of anvils without any change in the CRE of

individual DCCs.

Diagnosing a diurnal cycle related anvil cloud feedback in climate models may however be difficult. While Beydoun et al. (2021) found that changes in anvil lifetime contributed little to CRE feedbacks, this study used a radiative-convective-equilibrium model with no diurnal cycle of insolation. Although convective-resolving models have been found to model the diurnal cycle an lifecycle of DCCs better than parameterised climate models (Prein et al., 2015; Feng et al., 2023), but lack good observational constraints. Disentangling the impacts of convective processes and anvil cirrus processes on anvil lifecycle and CRE is also a key challenge. Here, the use of model experiments such as that of Gasparini et al. (2022) may help better understand the impacts of both processes on anvil CRE and the potential for climate feedbacks."

**Specific Comments**
*Line 161 "we only detect and track the thick portion of the anvil in this article": Can you be more specific about what "thick portion" means? For example, can you state the minimum cloud visible optical thickness that can be tracked by the algorithm?*

We have conducted a number of idealized, 1D radiative transfer simulations using libRadtran to assess the sensitivity of SEVIRI to anvil clouds at different heights and optical depths. We find that, using the detection thresholds in this study, the thick anvil detection is sensitive to optical depths between 1 and 1.5. This is backed up by the median of the minimum retrieved optical depth of tracked anvils of 1.45, and this value is likely high due to the inability to accurately retrieve OD at nighttime when many anvils dissipate. While this captures most of the CRE of anvil clouds, we agree that all CRE values are likely biased low by the inability to detect thin anvils. The following paragraph has been updated:

"Due to the lack of sensitivity of the SEVIRI SWD to thin ice clouds, we only detect and track the thick portion of the anvil in this article. The WVD channel of SEVIRI is capable of detecting anvils with optical thicknesses of approximately 1–1.5 (see supplementary fig. S1). However, the closer spacing and narrower bandwidth of the SEVIRI LW window channels (see supplementary fig. S2), along with the higher noise means that the SWD is less sensitive to thin cirrus compared to instruments such as the GOES-16 ABI (see supplementary fig. S3). The anvils tracked in this paper have a median retrieved minimum optical depth of 1.45, although this value is likely biased high as many anvils dissipate at night when accurate satellite retrievals of optical depth are not available. While this sensitivity captures much of the CRE of DCC anvils (Berry and Mace, 2014) the long lifetimes of dissipating thin anvils may have a significant warming contribution to net anvil CRE (Horner and Gryspeerdt, 2023). As a result, it is expected that the anvil CRE measured in this study are biased low."

In addition, a number of supplementary figures have been added, which are reproduced below:

[Figure]

**Figure S1.** Simulated sensitivity of the SEVIRI 10.8 μm BT (top) and WVD minus SWD (bottom) to anvil clouds of varying optical thickness at heights of 10, 12 and 14 km. The LibRadTran model was used to estimate the observed radiances, and all simulations used ice clouds with cloud top particle effective radius of 20 μm. The grey dashed line shows the 241 K BT, which, although commonly used as a threshold for anvil detection in satellite imagery, shows large sensitivity of the minimum optical thickness detected with the height of the anvil cloud. The grey region in the lower plot shows the range of temperatures in which the edge of the anvil is detected, as described in Jones et al. (2023). Similar sensitivity is found for all three cloud heights, with the optical depths of around 1–1.5 seen in the middle of the hysteresis region. The median minimum retrieved optical depth of all tracked anvils in our dataset is found to be 1.45, although this value is biased high by the inability to retrieve optical depth accurately at night-time.

[Figure]

**Figure S2.** Comparison of the relative spectral response (RSR) functions for the GOES-16 ABI and Meteosat-11 SEVIRI thermal IR channels. The LW window channels on ABI (channels 13 and 15) have a wider spacing than those of SEVIRI (channels IR10.8 and IR12.0). This wider spacing allows ABI to be more sensitive to the emissivity difference of ice clouds at wavelengths between 10 and 12 μm, and so it is better able to detect thin cirrus clouds.

[Figure]

**Figure S3.** Comparison of the sensitivities of ABI (dashed lines) and SEVIRI (solid lines) to anvil clouds of different optical thickness, using the LibRadTran simulation of an anvil at 14 km as used in fig. S1. The 10.8 μm BT (top panel) and WVD (middle panel) show very similar values for both instruments. The simulations of the SWD (bottom panel) show that SEVIRI is only about half as sensitive as ABI to thin ice clouds.

*Section 4.2: I think the current analysis in this section is well done, but I wonder if an even stronger signal would emerge if the analysis was performed separately with land-based convection and ocean-based convection. I think that oceanic clouds are typically larger, longer lasting, and have less intense convection than land-based clouds, so the land-ocean contrast may alias into the statistics in Fig. 6 and Fig. 7.*

While we considered investigating land-sea differences in this study, the domain area covering mostly land combined with the time range in which the ITCZ is at its Northernmost extent meant that only a small proportion of detected DCCs were over the sea (11%). As a result, we decided that this amount was too small for meaningful analysis. We have added the following sentence to the manuscript to make this clear:

"While the studied domain contains both land and sea regions, only a small proportion of tracked DCCs occurred over sea (11%), and so we have not separated the analysis of land and oceanic DCCs in this article."

We are currently performing a subsequent study over a larger domain in which we are investigating land-sea contrasts in anvil CRE.

*Line 284: This paragraph is written in a way that seems to imply that the average anvil-cloud net CRE must remain near zero as the climate changes. I'm not aware of any convincing physical mechanism or conservation law that would require the net anvil-cloud*

*CRE to remain near zero. Can you please explain why you think it will remain near zero or acknowledge the possibility that it will not remain near zero?*

The paragraph was not intended to apply that, but rather how shifts in the diurnal cycle could oppose anvil CRE feedbacks. The paragraph has been reworded to better discuss how changes in the diurnal cycle of convection could affect anvil CRE, without implying that this is part of a restoring mechanism. Please see the following updated paragraphs:

"It is apparent from figs. 11 and 12 that the observed neutral net anvil CRE is not only due to a balance between the SW and LW, but also from a balance of the cooling effect of daytime DCCs and the warming effect of those occurring at night. If the number of DCCs occurring during the daytime were to reduce we would expect a net warming effect without any change to the CRE of individual DCCs. As the diurnal cycle of convection over the ocean is nearly uniform, we should expect little impact on anvil CRE from changes in the time of convective initiation. However, over land, where convective activity is much more common in the afternoon, changes in the diurnal cycle may have a much larger effect on anvil CRE.

Furthermore, fig. 13 b highlights that differences in anvil temperature are linked to the diurnal cycle of anvil CRE as colder anvils tend to have longer lifetimes. As a result, if warming surface temperatures lead to the invigoration of DCCs, the warming effect we would see would be larger than the LW effect from the change in anvil temperature alone. Surface warming may also result in an earlier time of convective initiation, resulting in a cooling feedback."

**Technical Corrections**
*Line 15: The word "distribution" is used twice in the sentence. Consider changing to "We find that the anvil cloud CRE of our tracked DCCs has a bimodal distribution."*

Corrected

*Line 227 (and elsewhere): I think the name "Genio" should be "Del Genio"*

This was due to an error in the bibtex, and has been corrected throughout

*Line 279 "We see that, as expected, mean anvil CRE becomes more positive with increasing CTT": Should this be "mean anvil CRE becomes less positive or more negative …"*

Yes, corrected

*Line 308: change "outsize" to "outsized"*

Corrected

Additional references:

Berry, E. and Mace, G. G.: Cloud Properties and Radiative Effects of the Asian Summer Monsoon Derived from A-Train Data, Journal of Geophysical Research: Atmospheres, 119, 9492–9508, https://doi.org/10.1002/2014JD021458, 2014.

Horner, G. and Gryspeerdt, E.: The Evolution of Deep Convective Systems and Their Associated Cirrus Outflows, Atmospheric Chemistry and Physics, 23, 14 239–14 253, https://doi.org/10.5194/acp-23-14239-2023, 2023

Gasparini, B., Sokol, A. B., Wall, C. J., Hartmann, D. L., and Blossey, P. N.: Diurnal Differences in Tropical Maritime Anvil Cloud Evolution, Journal of Climate, 35, 1655–1677, https://doi.org/10.1175/JCLI-D-21-0211.1, 2022.

Prein, A. F., Langhans, W., Fosser, G., Ferrone, A., Ban, N., Goergen, K., Keller, M., Tölle, M., Gutjahr, O., Feser, F., Brisson, E., Kollet, S., Schmidli, J., van Lipzig, N. P. M., and Leung, R.: A Review on Regional Convection-Permitting Climate Modeling: Demonstrations, Prospects, and Challenges, Reviews of Geophysics, 53, 323–361, https://doi.org/10.1002/2014RG000475, 2015.

Feng, Z., Leung, L. R., Hardin, J., Terai, C. R., Song, F., and Caldwell, P.: Mesoscale Convective Systems in DYAMOND Global Convection-Permitting Simulations, Geophysical Research Letters, 50, e2022GL102 603, https://doi.org/10.1029/2022GL102603, 2023

Beydoun, H., Caldwell, P. M., Hannah, W. M., and Donahue, A. S.: Dissecting Anvil Cloud Response to Sea Surface Warming, Geophysical Research Letters, 48, e2021GL094 049, https://doi.org/10.1029/2021GL094049, 2021

Seidel, S. D. and Yang, D.: Temperatures of Anvil Clouds and Radiative Tropopause in a Wide Array of Cloud-Resolving Simulations, Journal of Climate, 35, 8065–8078, https://doi.org/10.1175/JCLI-D-21-0962.1, 2022.

---

## Author Comment (AC3)

There have been several minor changes made to the manuscript to address issues not raised by either reviewer. These are as follows:

1. Recalculation of CRE bias using updated CERES EBAF Ed 4.2 data after an error in the fluxes over certain scenes was corrected. The new bias is calculated as -1.87 (new), -2.02 (SW) and +0.15 (LW). This results in a small change to the net anvil CRE calculate at $-0.94\pm0.91Wm^{-2}$. This change does not affect the findings of the paper overall.
2. Update of the figures to better meet ACP guidelines.
3. Addition of a supplementary figures file to show greater detail on some technical aspects of the article.